# Uphill energy transfer mechanism for photosynthesis in an Antarctic alga

Makiko Kosugi [1,2,3,15,16] ✉, Masato Kawasaki[4,5,16], Yutaka Shibata [6,16] ✉, Kojiro Hara [7], Shinichi Takaichi [8], Toshio Moriya [4], Naruhiko Adachi [4], Yasuhiro Kamei [9,10], Yasuhiro Kashino [11], Sakae Kudoh [12,13], Hiroyuki Koike [3] & Toshiya Senda [4,5,14] ✉

*Prasiola crispa*, an aerial green alga, forms layered colonies under the severe terrestrial conditions of Antarctica. Since only far-red light is available at a deep layer of the colony, *P. crispa* has evolved a molecular system for photosystem II (PSII) excitation using far-red light with uphill energy transfer. However, the molecular basis underlying this system remains elusive. Here, we purified a light-harvesting chlorophyll (Chl)-binding protein complex from *P. crispa* (Pc-frLHC) that excites PSII with far-red light and revealed its ring-shaped structure with undecameric 11-fold symmetry at 3.13 Å resolution. The primary structure suggests that Pc-frLHC evolved from LHCI rather than LHCII. The circular arrangement of the Pc-frLHC subunits is unique among eukaryote LHCs and forms unprecedented Chl pentamers at every subunit–subunit interface near the excitation energy exit sites. The Chl pentamers probably contribute to far-red light absorption. Pc-frLHC's unique Chl arrangement likely promotes PSII excitation with entropy-driven uphill excitation energy transfer.

The capture of light energy and its transfer to photosynthetic reaction centers are the primary photosynthetic processes performed in light-harvesting proteins with photosynthetic pigments such as chlorophylls and carotenoids. To adapt to the various light conditions that exist in diverse micro-environments on Earth, pigmented species and their binding proteins became highly diversified during their evolution[1,2].

Eukaryotic photoautotrophs commonly use chlorophyll *a* (Chl *a*)-based light-harvesting complexes (LHCs)[2,3]. Subunits of LHCs each

[1]Astrobiology Center, 2-21-1 Osawa, Mitaka, Tokyo, 181-8588, Japan. [2]National Astronomical Observatory of Japan, 2-21-1 Osawa, Mitaka, Tokyo, 181-8588, Japan. [3]Department of Biological Sciences, Faculty of Science and Engineering, Chuo University, 1-13-27 Kasuga, Bunkyo-ku, Tokyo, 112-8551, Japan. [4]Structural Biology Research Center, Photon Factory, Institute of Materials Structure Science, High Energy Accelerator Research Organization (KEK), 1-1 Oho, Tsukuba, Ibaraki, 305-0801, Japan. [5]Department of Materials Structure Science, School of High Energy Accelerator Science, The Graduate University of Advanced Studies (Soken-dai), 1-1 Oho, Tsukuba, Ibaraki, 305-0801, Japan. [6]Department of Chemistry, Graduate School of Science, Tohoku University, 6-3 Aza Aoba Aramaki, Aoba-ku, Sendai 980-8578, Japan. [7]Department of Biological Production, Akita Prefectural University, 241-438 Kaidobata-nishi, Shimoshinjo-nakano, Akita 010-0195, Japan. [8]Department of Molecular Microbiology, Tokyo University of Agriculture, 1-1 Sakuragaoka, Setagaya-ku, Tokyo, 156-8502, Japan. [9]National Institute for Basic Biology, National Institutes of Natural Sciences, 38 Nishigonaka, Myodaiji, Okazaki, Aichi, 444-8585, Japan. [10]Department of Basic Biology, School of Life Sciences, SOKENDAI (The Graduate University for Advanced Studies), 38 Nishigonaka, Myodaiji, Okazaki, Aichi, 444-8585, Japan. [11]Graduate School of Science, University of Hyogo, 3-2-1 Kohto, Kamigohri, Ako-gun, Hyogo 678-1297, Japan. [12]National Institute of Polar Research, Research Organization of Information and Systems, 10-3 Midori-cho, Tachikawa, Tokyo, 190-8518, Japan. [13]Department of Polar Science, School of Multidisciplinary Science, SOKENDAI (The Graduate University for Advanced Studies), 10-3 Midori-cho, Tachikawa, Tokyo, 190-8518, Japan. [14]Faculty of Pure and Applied Sciences, University of Tsukuba, 1-1-1 Tennodai, Tsukuba, Ibaraki, 305-8572, Japan. [15]Present address: National Institute for Basic Biology, National Institutes of Natural Science, 38 Nishigonaka, Myodaiji, Okazaki, Aichi, 444-8585, Japan. [16]These authors contributed equally: Makiko Kosugi, Masato Kawasaki, Yutaka Shibata. ✉e-mail: mkosugi@nibb.ac.jp; shibata@m.tohoku.ac.jp; toshiya.senda@kek.jp

typically have three transmembrane helices[4]. LHCs absorb mainly visible light and deliver the absorbed energy to photosystem I (PSI) or photosystem II (PSII)[5–7]. PSI requires excitation energy corresponding to 700 nm light[8], and the light-harvesting chlorophyll protein complexes of PSI (LHCI) use up to approximately 720 nm light for PSI excitation[9]. To absorb long-wavelength light, LHCIs have long-wavelength-absorbing Chls (LWC), which consist of Chl *a* dimers[10–12]. On the other hand, the light-harvesting chlorophyll protein complexes of PSII (LHCIIs) usually do not have LWC. PSII typically uses excitation energy of 680 nm light to split water molecules[13]. Since LHCII contains Chl *a* with a red absorption band (Qy band) around 670 nm, monomeric Chl *a* is efficiently used in downhill excitation energy transfer (EET) in LHCII. A shorter-wavelength-absorbing Chl, Chl *b*[14], is also utilized for EET. Therefore, effective excitation of the eukaryotic PSII reaction center with far-red light (700–800 nm) is rare, although some examples have been reported in algae[15–17].

*Prasiola crispa* is one of the dominant green algae in Antarctica (Supplementary Fig. 1a)[18]. *P. crispa* often makes large, layered colonies in terrestrial habitats. These habitats are exposed to the triple stresses of severe cold, drought, and strong sunlight. The cost of the protective strategies against these stresses causes a drastic decrease in net photosynthesis[19–21]. While cells beneath the layered colony can avoid physical damage to their photosynthetic proteins induced by the triple stresses, the amount of photons absorbed to cells considerably decrease. Photosynthetically active radiation (400–700 nm) declines in the deep layers and only far-red light remains. (Supplementary Fig. 1b). Our analysis using cells of *P. crispa* suggested that the far-red

light absorbed by LWC allows PSII activation with nearly the same efficiency as that by visible light absorption in normal Chls[22]. The LWC thus has a critical role in increasing photosynthetic productivity inside the layered colony (Supplementary Fig. 1c-e). Since PSII excitation requires light of approximately 680 nm, the far-red light-harvesting system requires the uphill EET for PSII. Our earlier in vivo study revealed that *P. crispa* achieves efficient uphill EET with far-red light at the same global quantum yield as downhill EET with visible light[22]; this yield is defined as the ratio of photons used for PSII excitation to total photons absorbed by thalli[23]. As the first step to clarify the mechanism underlying the uphill EET in *P. crispa*, we purified an LWC-containing complex from *P. crispa* and analyzed it both spectroscopically and structurally. We have unveiled the structural basis for the survival strategy of *P. crispa* in an extreme environment based on its efficient utilization of far-red light.

## Results
### Identification of an LWC-binding protein
We purified the LWC-binding protein complex from thylakoid membranes of *P. crispa* by sucrose density gradient centrifugation and performed anion exchange chromatography by monitoring the far-red absorption band (Fig. 1a). The purified LWC-binding protein complex was designated Pc-frLHC (*Prasiola crispa* far-red light-harvesting Chl-binding protein complex). In addition to a typical absorption band at 671 nm (Qy band), Pc-frLHC shows a large far-red absorption band at 706.5 together with fluorescence emission at around 713 nm (F713) at room temperature, but the PSI-LHCI, PSII-LHCII, and LHCII fractions

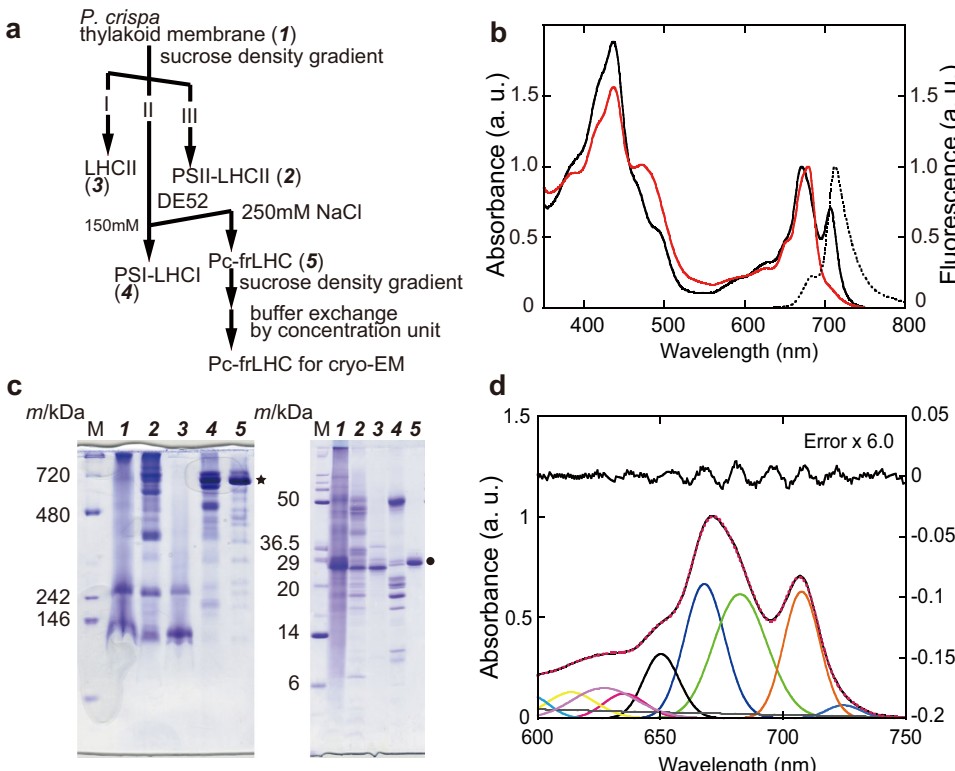

**Fig. 1 | Purification of Pc-frLHC. a** Purification scheme of Pc-frLHC and other photosynthetic proteins from *P. crispa*'s thylakoid membranes. **b** Absorbance spectra (solid lines) and fluorescence spectrum (dotted line) of the thylakoids (red) and Pc-frLHC (black) measured at room temperature. **c** hrCN-PAGE (left) and SDS-PAGE (right) analyses of thylakoids (lane 1), PSII-LHCII (lane 2), LHCII (lane 3), PSI-LHCI (lane 4), and Pc-frLHC (lane 5). Lane numbers at the top of the gel images correspond to the italicized numbers in parentheses in (**a**). Bands corresponding to Pc-frLHC and the subunit of Pc-frLHC are indicated by the star and the bullet, respectively. These electrophoresis data were representative of two independent

experiments. **d** Fitting analysis of the absorbance spectrum of purified Pc-frLHC at room temperature. The peak wavelength of each component was estimated by the second and fourth derivatives of the absorbance spectrum, and fitting analysis with Gaussian functions was performed by Magic plot 2.7.2 (Magicplot Systems, St. Petersburg, Russia). The components of $LWC_{708}$ and $LWC_{725}$ are shown by the orange and blue lines, respectively. The observed absorbance spectrum and the sum of each component of the Gaussian function are shown by black and dotted red lines, respectively. a.u.: arbitrary unit. Source data are provided as a Source Data file.

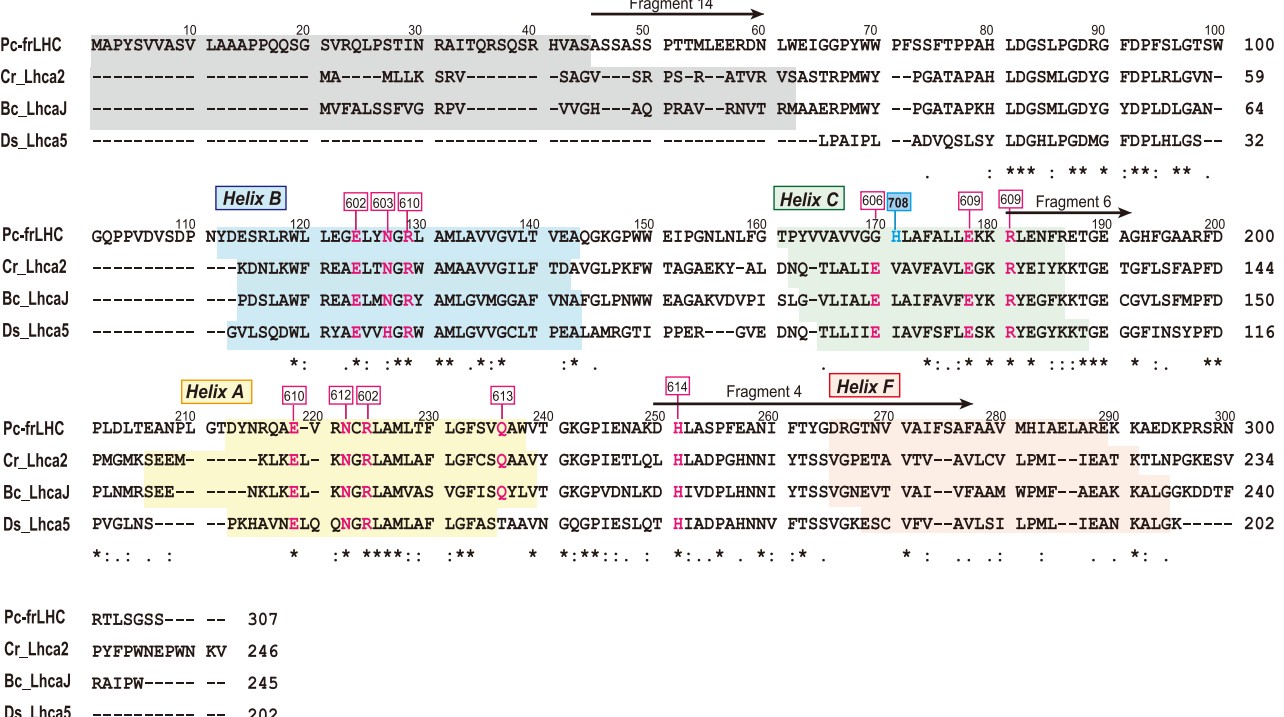

**Fig. 2 | Sequence alignment of Pc-frLHC from *P. crispa* and the closely related LHCs.** The deduced amino acid sequence of Pc-frLHC was classified into one of the LHCI groups along with Lhca2 of *Chlamydomonas reinhardtii* (Cr_Lhca2), Lhca-J of a marine green alga, *Bryopsis corticulans* (Bc_LhcaJ), and Lhca5 of a halophilic green alga, *Dunaliella salina* (Ds_Lhca5) with amino acid sequence identities of 32%, 32%, and 29%, respectively. Recent analyses revealed that Cr_Lhca2, Bc_LhcaJ, and Ds_Lhca5 are orthologs and loosely bound to the side of green algal PSI in a heterodimeric state together with Cr_Lhca9, Bc_LhcI, and Ds_Lhca6, respectively[24–28]. On the basis of these results, we concluded that Pc-frLHC is phylogenetically a member of LHCI, while it transfers excitation energy to PSII. Transmembrane helices are shaded in different colors, the signal peptides are shaded, chlorophyll-binding sites (followed by the nomenclature of Liu et al[32].) are shown as red characters, and the sequences detected by N-terminal sequences are shown by arrows (The $Mg^{2+}$ in Chl606 indirectly interact with Glu via a water molecule). The signal peptides and the transmembrane helices were predicted from the 3D structures registered in the PDB (Cr_Lhca2; 6JO5, 6IJO, Bc_LhcaJ; 6IGZ, Ds_Lhca5; 6SL5) and from the results of secondary structural prediction using Jpred 4[48] and TargetP-2.0[49]. The Chl708 binding site in *P. crispa* is shown in blue.

did not have a clear far-red absorbance except in the case of the PSI-LHCI complex with a small fraction of LWC (Fig. 1b and Supplementary Fig. 2). Pc-frLHC is a homo-oligomer composed of a subunit of about 29 kDa (Fig. 1c, right). The band of Pc-frLHC appeared at approximately the same position as that of the PSI-LHCI super complex and the 700 kDa marker protein on the native PAGE (Fig. 1c, left). Partially decomposed complexes are detected as dilute bands in the lower molecular side of the complete oligomer. A fitting analysis of the absorbance spectrum at room temperature was performed, revealing that the far-red Qy band fitted by two components peaking at 708 and 725 nm accounted for about 22% and 2% of the total Qy band including Chl *b*, respectively. The area ratio of the far-red Qy bands of the two LWC components peaking at 708 ($LWC_{708}$) and 725 nm ($LWC_{725}$) was 11:1 (Fig. 1d).

The amino acid sequence of the subunit of Pc-frLHC was deduced from cDNA sequences of the total mRNA libraries with the help of an internal amino acid sequence analysis of Pc-frLHC (Fig. 2 and Supplementary Fig. 3). The LHCII family proteins are known to be the only LHCs that contribute to PSII excitation in the green lineage of photosynthetic eukaryotes (green algae and plants). Surprisingly, while Pc-frLHC delivers the excitation energy to PSII, Pc-frLHC is not a member of the LHCII family. The amino acid sequence of the Pc-frLHC subunit was classified into one of the green algal LHCI groups that each have four transmembrane helices, such as Lhca2 of *Chlamydomonas reinhardtii* (Cr_Lhca2)[24–26], Lhca-j of *Bryopsis corticulans* (Bc_LhcaJ)[27], and Lhca5 of *Dunaliella salina* (Ds_Lhca5)[28], whereas other LHCI groups each have three transmembrane helices (Fig. 3). Cr_Lhca2, Bc_LhcaJ, and Ds_Lhca5 are synonymous with each other and bind at a side of PsaI in PSI.

## Structure of Pc-frLHC

Using cryogenic electron microscopy (cryo-EM), the structure of Pc-frLHC was determined at 3.13 Å resolution (Fig. 4, Supplementary Table 1 and Supplementary Fig. 4). Pc-frLHC is an undecamer with 11-fold symmetry (Fig. 5a). Ring-shaped light-harvesting antennas have been reported only from LH1 and LH2 in anaerobic purple bacteria and from IsiA in cyanobacteria[29–31]. Pc-frLHC is the first example of a ring-shaped eukaryotic LHC. The stromal and lumenal sides of Pc-frLHC were determined by referring to the common tertiary structures of eukaryotic LHCs, in which the sidedness of the $NH_2$ terminus and the arrangement of helices B, C, and A from the N-terminus are conserved[4–6](Fig. 5b). However, the subunit of Pc-frLHC has three unique structural features. First, the N-terminal loop region is significantly longer than its homologs (Fig. 2). Second, the BC loop is relatively short and lacks a well-conserved helix-E. Third, the subunit of Pc-frLHC has a fourth transmembrane helix, helix-F, after helix-D, as observed in LHCI-type homologs of Pc-frLHC (Figs. 2, 5b and 6)[24–28]. The superposition of the Pc-frLHC subunit and Cr_Lhca2 showed a difference in the arrangement of helix-F relative to the other transmembrane helices[26] (Fig. 6).

Among the structurally known LHCs, subunit–subunit interactions in the ring structure of Pc-frLHC are most similar to those between Lhca1a and Lhca1b subunits in the halfmoon-shaped belt of *C. reinhardtii* LHCI[26]. However, there is a significant difference between them. The interactions between Lhca1a and Lhca1b are reinforced by residues of another subunit, Lhca8 (Fig. 6). On the other hand, subunit–subunit interactions in Pc-frLHC are strengthened by the long N-terminal regions of the subunit. The long N-terminal region of every Pc-frLHC subunit interacts with the N-terminal regions of both sides of

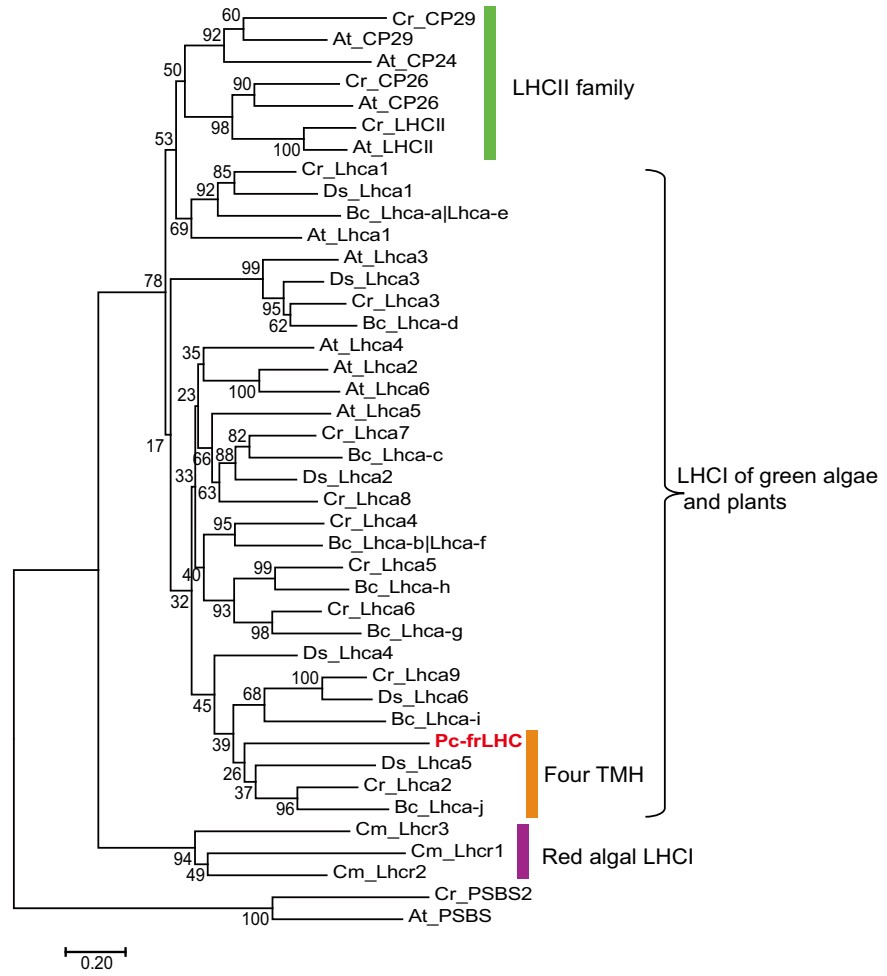

**Fig. 3 | Phylogenetic tree of light-harvesting complexes (LHCs) of PSI and PSII in green algae and plants.** Evolutionary analyses of LHCs were conducted with the neighbor-joining method in MEGA7[50]. We used amino acid sequences of only those LHCs in which 3D structural analyses had been achieved. The optimal tree with the sum of branch lengths = 17.75479796 is shown. The percentage of replicate trees in which the associated taxa clustered together in the bootstrap test (1000 replicates) is shown next to the branches. The evolutionary distances were computed using the Poisson correction method. The rate variation among sites was modeled with a gamma distribution (shape parameter = 3). While most LHC subunits have only three transmembrane helices (TMH), a few LHC subunits, such as Cr_Lhca2, Bc_LhcaJ (Bc_Lhca-j in the figure), and Ds_Lhca5, have been known to possess a fourth transmembrane helix. Abbreviations: Cm: *Cyanidioschyzon merolae* (red alga); Cr: *Chlamydomonas reinhardtii*; Bc: *Bryopsis corticulans*; Ds: *Dunaliella salina*;

Pc: *Prasiola crispa* (green algae); At: *Arabidopsis thaliana* (plant). Blast accession numbers: Cm_Lhcr1 (XP_005538084), Cm_Lhcr2 (XP_005537362), Cm_Lhcr3 (5ZGB_3), Cr_Lhca1 (6IJJ_1), Cr_Lhca2 (XP_001691031), Cr_Lhca3 (PNW76422), Cr_Lhca4 (6IJJ_4), Cr_Lhca5 (6IJJ_5), Cr_Lhca6 (6IJJ_6), Cr_Lhca7 (AAO16495), Cr_Lhca8 (6IJJ_8), Cr_Lhca9 (XP_001692548), Cr_CP29 (XP_001697193), Cr_PSBS2 (XP_001689923.1), Cr_CP26 (XP_001695927), Cr_LHCII (XP_001700243.1), Bc_Lhca-j (6IGZ_0) (Bc_LhcaJ), Bc_Lhca-a (6IGZ_1), Bc_Lhca-c (6IGZ_2), Bc_Lhca-d (6IGZ_3), Bc_Lhca-b (6IGZ_4), Bc_Lhca-g (6IGZ_6), Bc_Lhca-h (6IGZ_7), Bc_Lhca-i (6IGZ_9), At_Lhca1 (NP_191049.1), At_Lhca2 (NP_191706.2), At_Lhca3 (NP_001185280.1), At_Lhca4 (NP_190331.3), At_Lhca5 (NP_175137.1), At_Lhca6 (NP_173349.1), At_CP26 (NP_192772.1), At_CP29 (NP_195773.1), At_CP24 (NP_173034.1), At_LHCII (NP_174286.1), At_PSBS (NP_001319163.1), Ds_Lhca1 (6RHZ_1), Ds_Lhca2 (6RHZ_2), Ds_Lhca3 (6SL5_3), Ds_Lhca4 (6QPH_4), Ds_Lhca5 (6SL5_5), Ds_Lhca6 (6SL5_6).

the subunits in the ring-shaped structure (Fig. 6). The fourth transmembrane helix of Pc-frLHC, helix-F, is not involved in the subunit–subunit interaction.

While protein–protein interactions between subunits are abundant at the stromal side, they are limited at the lumenal side. As a result, there is a large cavity at the interface of the two subunits. The cavity accommodates eight Chl molecules (Chl601, 603, 609, 708, 611', 612', 613', 614') (Fig. 7; the residue number of Chl in a different subunit is shown with a prime). These Chls seem to support interactions between subunits. We recognized ~60% of the Pc-frLHC particle images that contained unidentified components in the ring's hole (approximately 120 Å diameter), although their structures could not be resolved by single particle analysis with C11 symmetry. While we have tried single particle analysis without a symmetry assumption, no interpretable cryo-EM density was obtained in the central hole. These components filling the central hole might be detergent, because the

SDS-PAGE analysis showed few proteins other than Pc-frLHC (Fig. 1c). Since the surface of the hole is lined mainly by hydrophobic residues, it seems possible that it interacts with detergent. PSII in *P. crispa* is expected to be loosely bound outside of the Pc-frLHC ring structure, because neither the PSI subunit nor the PSII subunit was detected by SDS-PAGE of the Pc-frLHC fraction. Moreover, PSII seems to be too large to fit into the inner space of the ring.

**Pigment arrangement in Pc-frLHC**

We found 11 Chls and two carotenoids in the subunit of Pc-frLHC (Fig. 5c and Supplementary Table 2). While high-performance liquid chromatography (HPLC) pigment analysis using a C18 column suggested that at least one Chl in the subunit is Chl *b* (Supplementary Fig. 5), it was impossible to distinguish Chl *a* and *b* in the current cryo-EM map. We, therefore, modeled the 11 Chls as Chl *a*. The numbering of Chls in Pc-frLHC was defined following the nomenclature of spinach

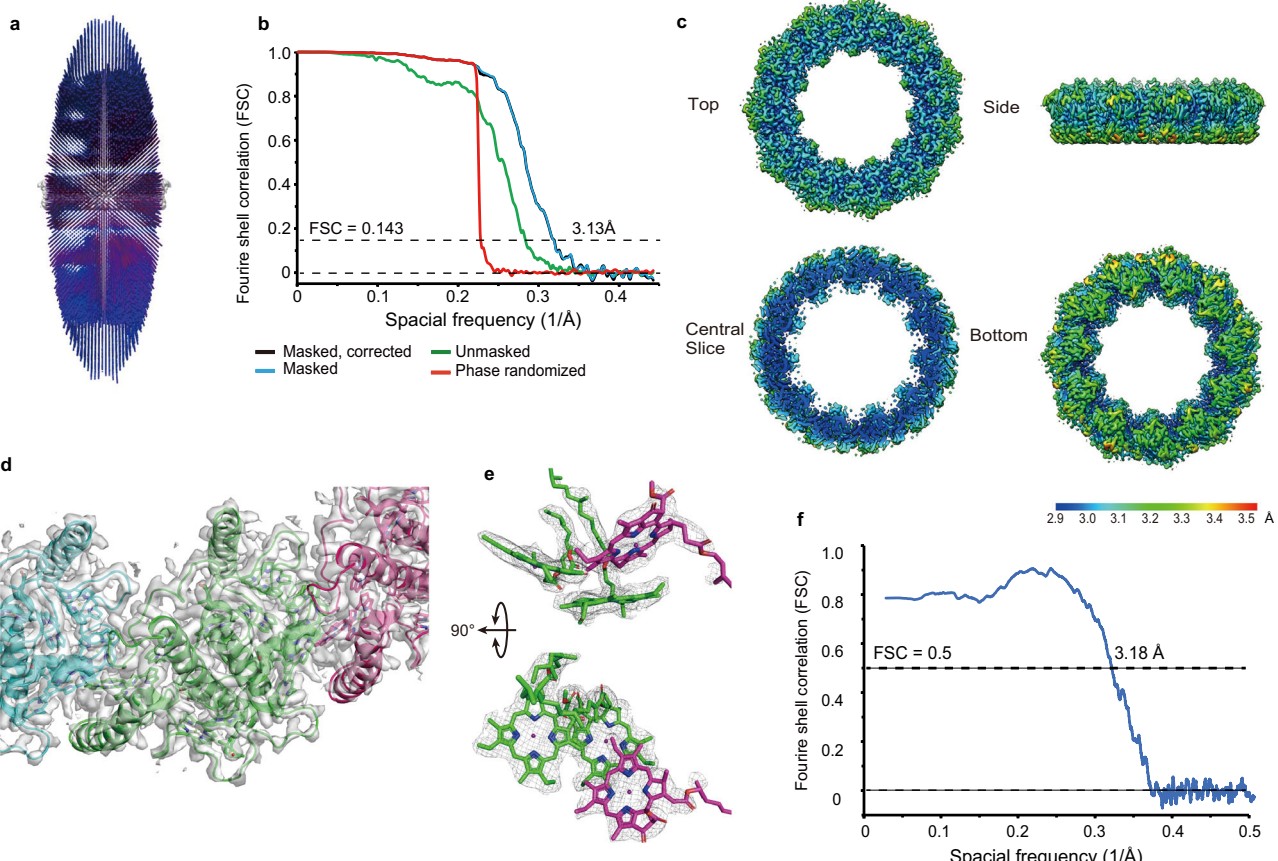

**Fig. 4 | Summary of Cryo-EM analysis. a** Angular distribution of the cryo-EM particles. **b** Gold standard Fourier shell correlation (FSC) curves of the refined 3D reconstruction. **c** The 3D reconstruction is colored according to the local resolution. Top: stromal side; bottom: lumenal side. **d** Structures of Pc-frLHC subunits fitted to the cryo-EM map. **e** Representative densities of the Chl trimer. **f:** Map-to-model FSC curve.

LHCII[32]. Except for Chl708, all Chls in the subunit occupy positions similar to those of the corresponding Chls in spinach LHCII monomers. Chls form two layers; one is close to the stromal side (Chls601, 602, 603, 609, 610, 611, and 612) and the other is close to the lumenal side (Chls604, 613, and 614) (Fig. 5c). Chl708 is located between the two layers.

Due to the subunit–subunit interactions, seven consecutive Chls on the stromal layer (Chl601, 602, 603, 609, 611', 612', 610') are aligned from inside to outside the undecameric ring, with Mg–Mg distances less than 15 Å (Fig. 7). The calculated exciton couplings (Supplementary Table 3) suggest that two sets of the seven consecutive Chls in neighboring subunits are energetically connected to each other through the Chl610–Chl602/Chl603 and Chl601–Chl603/Chl609 interactions (Fig. 7 and Supplementary Table 3). All Chls on the stromal layer form an energetically consecutive network throughout the undecameric ring. In addition, Chl708 energetically connects the stromal Chl603-609 dimer and Chl611' to the lumenal Chl613'-614' dimer.

Three carotenoid-binding sites, L1, L2 (both surrounding central helices A and B), and N1 (near helix C), are conserved among LHCs. Biochemical analysis shows that Pc-frLHC binds only two carotenoids: violaxanthin (vio) and loroxanthin (loro). The vio and loro in the all-trans configuration were assigned in the L1- and L2-binding sites of Pc-frLHC, respectively (Fig. 5c). The N1 site is not occupied in Pc-frLHC.

## Multimeric structure of Chls and their spectroscopic assignments

In the Pc-frLHC monomer, Chl603 and 609 were estimated to have the strongest exciton coupling among the Chls in Pc-frLHC

(Supplementary Table 3). Chl708 also has strong exciton couplings to both Chl603 and 609. Chl708 and 603 are located side by side, and Chl609 makes stacking (π–π) interactions with Chl708 and 603 (Fig. 7); Chl603, 609, and 708 form a unique trimeric structure. This trimer can be a strong candidate for the LWC since it is structurally similar to the Chl trimer in cyanobacterial PSI that has been assigned to an LWC[33,34]. In addition, Chl613' and Chl614' show strong exciton couplings with the trimeric Chls. Therefore, the LWC absorbance can be assigned to the delocalized excited state over the trimer, which possibly spreads to Chl613' and Chl614'. The area ratio of far-red Qy absorbance of LWCs in the purified Pc-frLHC sample was calculated to be 24%, i.e., as the sum of 22% for LWC$_{708}$ and 2% for LWC$_{725}$ (Fig. 1d). The absorbance of aggregated Chls seems to spread in a large wavelength area from red to far-red.

Of the seven consecutive Chls on the stromal layer, Chls610, 611, and 612 are likely to have a critical function; these Chls probably serve collectively as the energy exit site. Since these Chls are located near the outside of the Pc-frLHC ring (Figs. 5 and 7), they can interact with PSII on the outer surface of the Pc-frLHC ring structure. In structurally known LHCII–PSII complexes, the Chls for the energy exit are located at the interface between LHCII and PSII[6,35–37].

## Fluorescence measurement revealed two distinct red Chl *a* pools

To analyze the details of the EET of Pc-frLHC, we analyzed the spectroscopic properties of Pc-frLHC. First, we measured the time-resolved fluorescence spectrum at 273 K with excitation at 740 nm light to confirm the uphill energy transfer from the LWC to the bulk Chl emitting around 680 nm. The direct excitation of LWC by the 740 nm

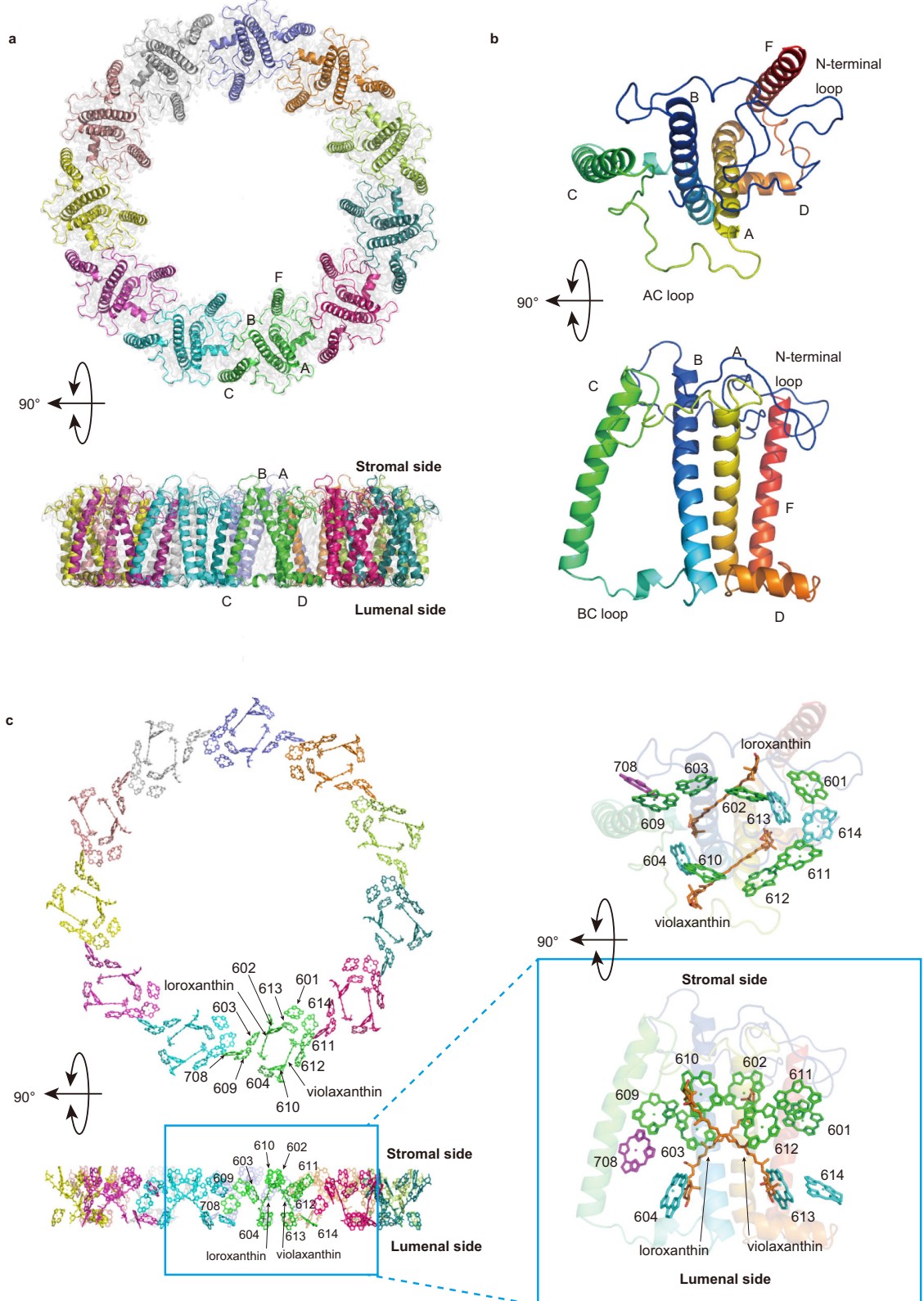

**Fig. 5 | Overall cryo-EM structure of Pc-frLHC. a** Top (upper panel) and side (lower panel) views from the stromal side and outside of the undecamer ring, respectively, of Pc-frLHC are shown with the cryo-EM map. Each subunit is shown in a different color. **b** Top and side views (the same views as in (**a**)) of the subunit of Pc-frLHC in rainbow colors from the N-terminus in blue to the C-terminus in red. **c** Chlorophyll *a* arrangement (the same views as in (**a**)) in Pc-frLHC (left panel). Each chlorophyll appears in the same color as the corresponding subunit in (**a**). The right panel shows the arrangement of chlorophyll *a* in the subunit. Chlorophylls on the stromal and lumenal sides are shown in green and cyan. Chl708 is shown in purple.

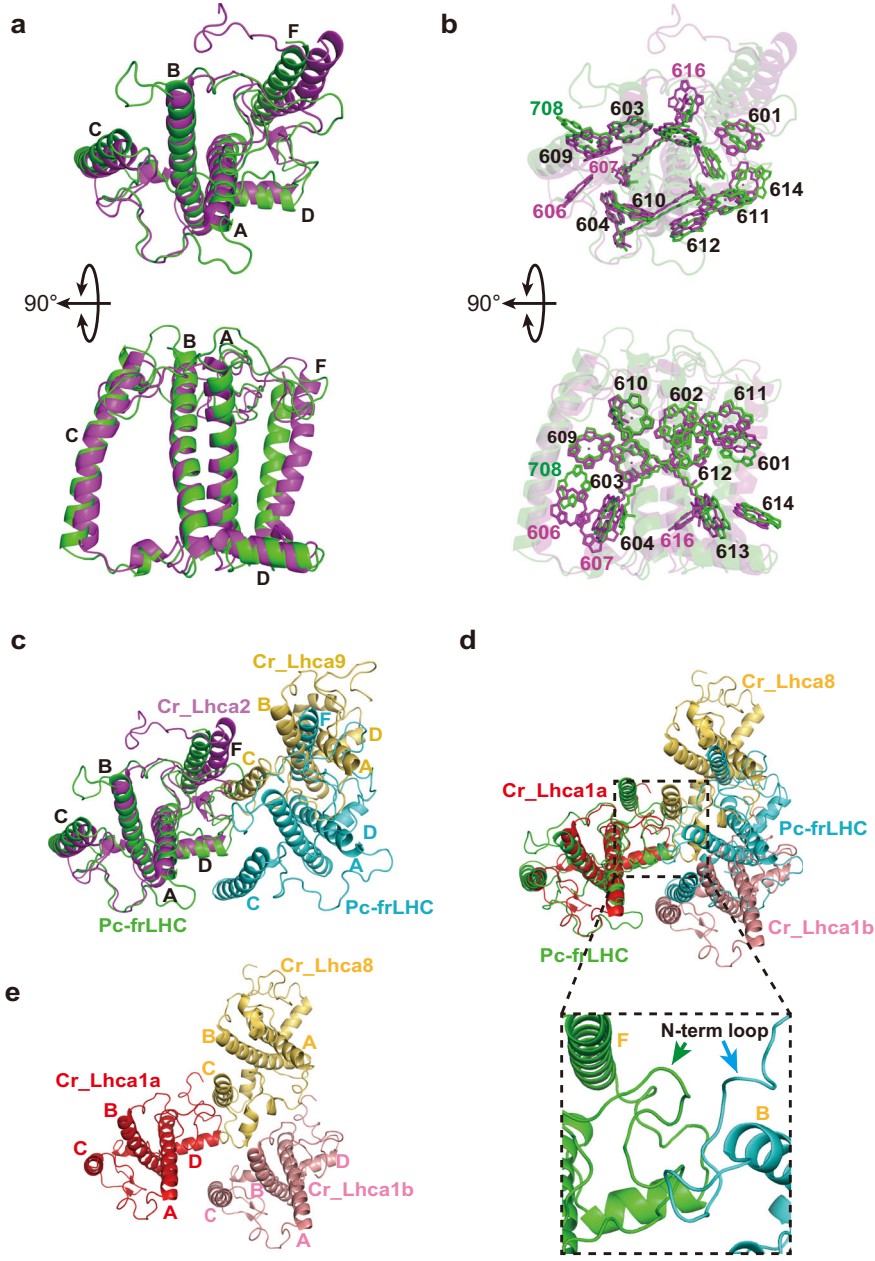

**Fig. 6 | Superposition of Pc-frLHC subunit (green) and Cr_Lhca2 (magenta; PDB: 7DZ7). a** Top and side views of Pc-frLHC and Cr_Lhca2 (only peptide chains) from the stromal side. Four transmembrane helices (A, B, C and F) and one amphiphilic helix (D) are labeled. **b** The same views as in (**a**) showing Chls and carotenoids. Chls common to both proteins are shown with black numbers. Chls unique to Pc-frLHC and Cr_Lhca2 are shown with green and magenta numbers, respectively. **c** Comparison of the two neighboring subunits of Pc-frLHC (green and cyan) with Cr_Lhca2-Cr_Lhca9 heterodimer (PDB ID: 7DZ7) (magenta and yellow). Pc-frLHC (green) is superposed with Cr_Lhca2 (magenta). **d** Subunit–subunit interactions of Cr_Lhca1a-Cr_Lhca1b-Cr_Lhca8 (red, salmon, and yellow) and Pc-frLHC (green and cyan). Pc-frLHC (green) is superposed with Cr_Lhca1a (red). The bottom panel is an enlarged view of subunit–subunit interactions of Pc-frLHC by long N-terminal loop regions from neighboring subunits. **e** Subunit–subunit interactions of Cr_Lhca1a-Cr_Lhca1b-Cr_Lhca8. Transmembrane helices are labeled.

laser generated fluorescence at 680 nm (F680) with a rise time of 25 ps (Fig. 8a). It should be noted that the time constant of 25 ps is given by 1/($k_{up}$ + $k_{down}$), where $k_{up}$/$k_{down}$ is the rate constant of the uphill/downhill energy transfer. The longest time constant of the fluorescence decay was 2.2 ns. This result clearly confirmed the efficient utilization of far-red light in Pc-frLHC realized by the uphill EET from LWC to bulk Chls.

Next, we analyzed the energy transfer pathways from Chl *b* and carotenoids by measuring the excitation spectrum in the range of 400 nm to 550 nm with the monitoring wavelength at 730 nm at 77 K; we evaluated the relative energy transfer efficiency from Chl *a*, Chl *b*, and carotenoid to the 730 nm–emitting species. The spectral range covers the absorption bands of the carotenoid and the Soret bands of Chl *a* and Chl *b*. We compared the excitation spectrum with the 1-transmittance spectrum (Supplementary Fig. 5). The good agreement between the profiles of the two spectra indicates that the efficiencies of energy transfer to the 730 nm–emitting species were similar between these components. Since the 460 nm absorbance of Chl *a* is quite small, Chl *b* and the two carotenoids are likely excited by the 460 nm laser. Then, the absorbed energies seem to be transferred to the 730 nm–emitting species through Chl *a*.

We then analyzed details of the spectroscopic properties of LWC. As shown in a previous study[22], the fluorescence emission peak of

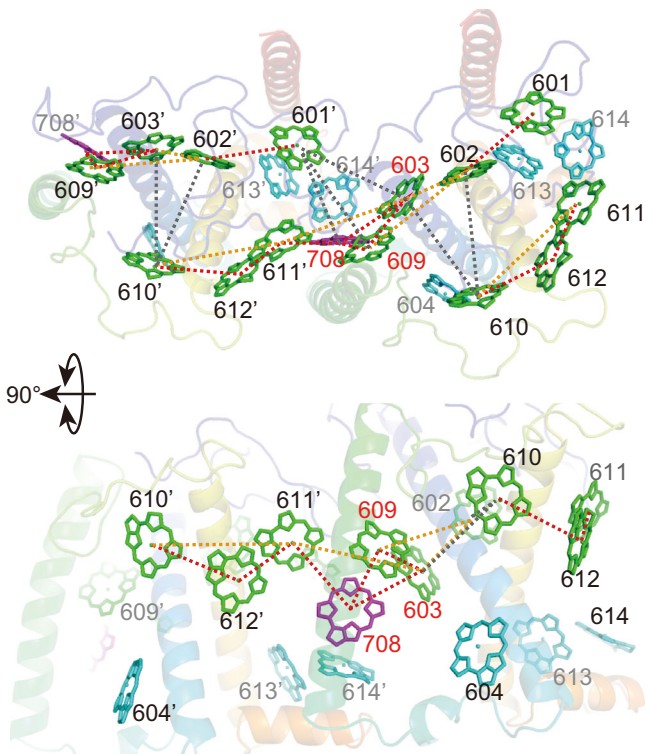

**Fig. 7 | The chlorophyll network in Pc-frLHC.** Connections among chlorophyll *a* in Pc-frLHC. Energetically connected stromal chlorophylls (including Chl708) are linked by dotted lines (Supplementary Table 3). The dotted lines are colored based on the values of exciton couplings (*EC* cm$^{-1}$ unit): *EC* > 60 in red, 60 ≥ *EC* > 30 in orange, and 30 ≥ *EC* > 10 in gray. Chlorophylls on the stromal and lumenal sides are shown in green and cyan, respectively. Chl708 is shown in purple. While Chls613' and 614' are energetically connected to Chls708, 612', and 601', for clarity they are shown only partially in this figure. The upper panel is viewed from the stromal side and the lower panel is viewed from outside the undecamer ring.

Pc–frLHC showed a temperature-dependent red shift from 713 nm (F713) at room temperature to 730 nm (F730) at 87 K (Fig. 8b). We found that the peak shift can be explained by assuming two intrinsic red Chl *a* pools emitting at 713 nm (assigned to LWC$_{708}$) and 730 nm (assigned to LWC$_{725}$); lowering the temperature alters the equilibrium between them, causing the red shift of the emission. A slight but significant broadening of the emission band in the intermediate temperature region from 144 K to 183 K suggests the overlap of the two emission bands, thus lending support to the above assumption. Based on this assumption, the fluorescence spectra at various temperatures were deconvoluted by the fitting of the two Gaussian functions with peaks at 713 nm and 730 nm (Fig. 8b). The relative amplitudes of the two peaks could be well explained by the Arrhenius relation (inset of Fig. 8b). Importantly, the above model requires that LWC$_{708}$ intrinsically emits stronger fluorescence than LWC$_{725}$. The ratio of the intrinsic emission intensity of F713 to that of F730 ($I_{F713}/I_{F730}$) was -12 (see the Methods section for details).

For further analysis, the time-resolved fluorescence at 80 K, 201 K, and 273 K was measured with excitation at 460 nm to avoid direct excitation of LWC (Chl *a*). In fact, Chl *b* and the carotenoids were excited mainly at 460 nm. The fluorescence decay curves were analyzed by the global exponential decay fitting to obtain the fluorescence decay–associated spectra (FDAS) shown in Fig. 8c-e. Typical experimental and fitting curves are shown in Supplementary Fig. 6. We obtained two FDAS components assigned to LWC at 80 K; one with a time constant of 250 ps peaking at around 710 nm and the other with a time constant of 1.9 ns peaking at around 730 nm. The former and

latter were assigned to F713 and F730, respectively. This result again supports the co-existence of two independent pigment pools. The FDAS analysis also revealed a decay of F713 and a rise of F730 with the same time constant of 250 ps (Fig. 8e), suggesting an energy transfer from the Chl *a* pool emitting F713 to that emitting F730 with a time constant of 250 ps.

We propose a plausible kinetic model for the EET dynamics in Pc-frLHC (Fig. 9) that comprehensively explains the temperature dependence of its fluorescence properties. EETs from the bulk Chl to both LWC$_{708}$ and LWC$_{725}$ take place with time constants of approximately 25 ps (Fig. 8a). The EET from LWC$_{708}$ to LWC$_{725}$ has a time constant of approximately 250 ps (Fig. 8e). As the temperature rises, the equilibrium between the two LWC pools shifts toward LWC$_{708}$ due to the severe temperature-dependent enhancement of the uphill EET. Because of the rapid exchange of the excitation energy between the two LWC pools, F713 and F730 decay with similar time constants. This results in the spectral merge of the two FDAS components at high temperatures. The temperature dependence of the ratio of the forward ($k_{708\rightarrow725}$) with respect to the backward ($k_{725\rightarrow708}$) uphill EET is expressed by Eq. 2 in the Methods section. The uphill EET rate $k_{725\rightarrow708}$ is largely enhanced by the great abundance of the pigment number constituting LWC$_{708}$ (entropic effect; see the Discussion section).

## Discussion

In this study, we purified Pc-frLHC and determined its structure at 3.13 Å resolution using cryo-EM, revealing a ring-shaped undecamer structure. The phylogenetic tree of amino acid sequences showed that Pc-frLHC is evolutionarily close to green algal four-transmembrane helices LHCI, such as Lhca2 of *Chlamydomonas reinhardtii* (Cr_Lhca2), Lhca-j of *Bryopsis corticulans* (Bc_LhcaJ), and Lhca5 of *Dunaliella salina* (Ds_Lhca5). Previous studies assigned the dimeric Chl603-609 in Cr_Lhca2 as the longest-wavelength Chls among all LHCI of *C. reinhardtii*[12]. Interestingly, Pc-frLHC has a trimeric Chl603-609-708 at the corresponding position. Chl708 is a unique Chl in Pc-frLHC, and a corresponding Chl has not been found in the other four-transmembrane helices LHCIs. Structural comparison between the subunit of Pc-frLHC and that of Cr_Lhca2 revealed that Chl606 in Cr_Lhca2 is spatially closest to Chl708 in Pc-frLHC. In Cr_Lhca2, a highly conserved Glu residue, Glu114, interacts indirectly with Mg$^{2+}$ of Chl606 through a water molecule; this interaction is also observed in Bc_LhcaJ and Ds_Lhca5. However, the conserved Glu is replaced by Gly170 in Pc-frLHC (Fig. 2). Instead, Pc-frLHC has His171, and this His residue coordinates directly the Mg$^{2+}$ of Chl708. This change in the ligand residue makes Chl708 close to Chl603-609, thus enabling the formation of the Chl603-609-708 trimer. Moreover, the ring-shaped arrangement of 11 subunits in Pc-frLHC enables Chl708 to interact with the Chl613'-614' dimer in the adjacent subunit, forming a Chl pentamer that may contribute to further red shift absorption.

We have demonstrated that Pc-frLHC in *P. crispa* can excite PSII with far-red light using uphill energy transfer from LWC to bulk Chls[22]. Pc-frLHC has two LWCs, a main far-red light-absorbing LWC, LWC$_{708}$, and a further red-shifted LWC, LWC$_{725}$. However, the structural studies suggested that the trimeric Chls603-609-708 is the only candidate for the far-red absorbing LWC, which might be spread over the neighboring dimeric Chls613'-614' via the strong exciton coupling. Since Chl613' and 614' are coordinated by ligand residues of the neighboring subunit harboring the trimer Chls, the formation of five spatially continuous Chls, which are composed of the trimer Chl and Chl613'-614', is possible by the ring-shaped structure of Pc-frLHC. Our spectroscopic analysis showed that there are two fluorescence species, F713 and F730. While it is difficult to provide a definite explanation for the ratio of $I_{F713}/I_{F730}$ ( = 12) calculated from our experimental results, one possible and simple explanation is that the 11 LWC sites can be divided into two species, LWC$_{708}$ and LWC$_{725}$. Therefore, we assume that a slight symmetry breakdown that could not be detected in the

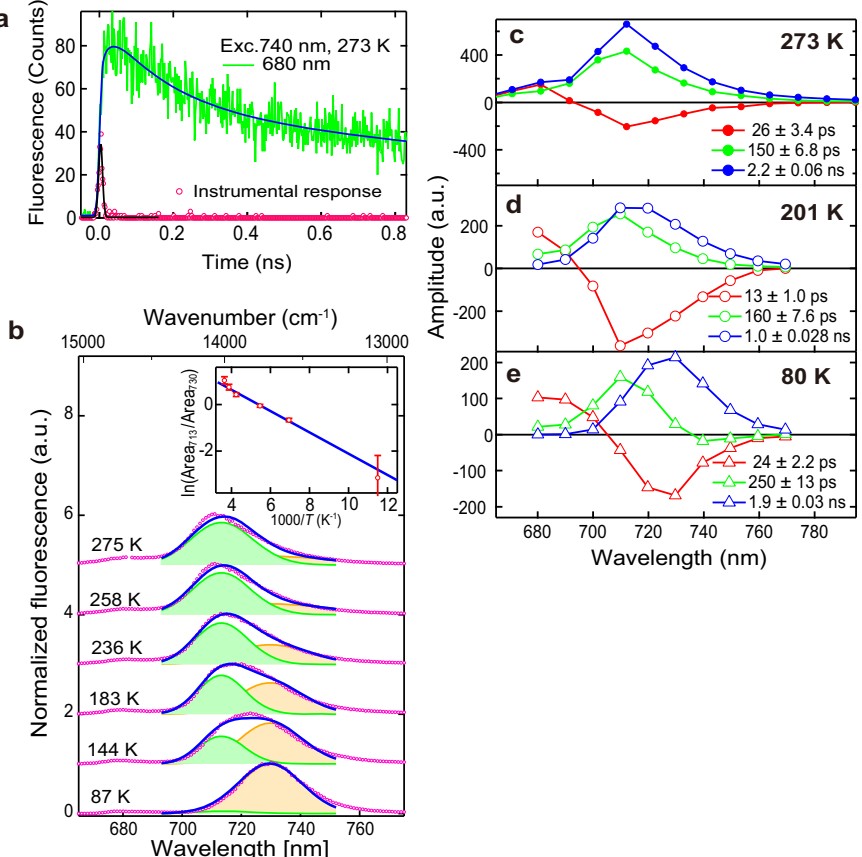

**Fig. 8 | Spectroscopic analysis of Pc-frLHC. a** Fluorescence time profiles of Pc-frLHC excited at 740 nm and monitored at 680 nm (green) observed at 273 K. The blue line shows the fitting curve to the sum of three exponential components convolved with the instrumental response function shown by red circles.
**b** Temperature dependence of the fluorescence spectrum of Pc-frLHC excited at 460 nm. The blue lines are the fitting curves to the sum of two Gaussian functions (the filled green and orange curves). The inset shows the analysis of the $\text{Area}_{F713}/$

$\text{Area}_{F730}$ ratio using the Arrhenius equation. The blue line is the fitting according to Eq. 2 (see Methods) with the energy gap fixed to 318 cm$^{-1}$. The error bars on the data in panel b inset were estimated from the standard errors of the fitting of the spectra to the sum of two Gaussian curves. **c–e** Fluorescence decay–associated spectra of Pc-frLHC excited at 460 nm and observed at 273 K (**c**), 201 K (**d**), and 80 K (**e**). Source data are provided as a Source Data file.

cryo-EM analysis induces a slight red shift for one of the 11 LWCs, giving rise to LWC$_{725}$. In this case, the pigment number ratio of LWC$_{708}$ with respect to LWC$_{725}$ is 10, which is in rough agreement with the value of 12 for the experimentally determined value of $I_{F713}/I_{F730}$.

Strongly coupled pigments form mixed excited states that can be either super-radiative or optically forbidden. Therefore, the large $I_{F713}/I_{F730}$ ratio would also be explained on the basis of the enhanced/reduced oscillator strengths of these Chl pools. To test whether this explanation is possible, we estimated the oscillator strengths of mixed excited states formed by Chl pairs (hatched with magenta in Supplementary Table 3) with significantly strong excitonic couplings. To appropriately explain the factor of 12 for $I_{F713}/I_{F730}$, the transition dipole moment of the lower mixed state should be reduced to ca. 30% of that of the higher mixed state. However, we could not find any Chl pairs that have satisfactorily enhanced/reduced oscillator strength to explain the large $I_{F713}/I_{F730}$ ratio (Supplementary Table 4). Therefore, we conclude here that the large $I_{F713}/I_{F730}$ ratio is attributable to the large Chl number of the F713 pool.

Due to the uphill EET, theoretically only part of the absorbed energy can be used for PSII excitation. However, our in vivo analysis of *P. crispa* suggested that the far-red light absorbed by LWC allows effective PSII activation with approximately the same frequency as that of visible light absorbed by normal Chls[22]. The highly efficient uphill EET from LWC$_{708}$ to bulk Chls in Pc-frLHC seems to be explained by the thermal activation and entropy effect. When the energy gap is 28 nm

(from 708 nm to 680 nm), 5% of the total population can be calculated in the excited state by the Boltzmann distribution at 278 K. Importantly, the estimated abundance ratio of the bulk Chls to Chls in LWC$_{708}$ in Pc-frLHC seems to increase the probability. The fitting analysis of the Qy-band absorption suggested that 76% and 24% of the absorption can be attributed to the bulk Chls and LWC, respectively. Therefore, the abundance ratio of bulk Chls to LWC$_{708}$ can be estimated as approximately 3:1, leading to a 15% probability of excitation (Fig. 9). Importantly, the ring-shaped structure of Pc-frLHC can significantly increase the probability of energy transfer to PSII. Since the ring-shaped structure forms an energetically connected Chl network (Fig. 7 and Supplementary Table 3), the connection between Pc-frLHC and PSII is equally possible from any of the 11 binding sites of Pc-frLHC, resulting in sufficient probability of PSII excitation *per LHC* with the far-red light. Taken together, we can conclude that the energetically continuous Chl $a$ network in a subunit contributes to the highly efficient EET of Pc-frLHC by the entropy effect (even in the monomeric Pc-frLHC). Moreover, the ring-shaped Chl $a$ network in the Pc-frLHC undecamer would provide another entropy effect for higher EET efficiency to PSII. Since Chls610, 611, and 612 are together located just beside a probable LWC of the next subunit, the excited energy from the LWC can be readily transferred to PSII.

While the highly efficient uphill EET can be explained as above, future tasks remain with respect to the assignment of Chl $b$ and the function of LWC$_{725}$ and carotenoids. First, we consider the function of

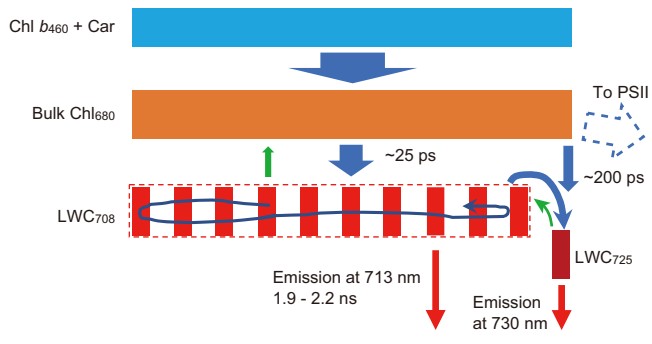

**Fig. 9 | Kinetic model of the EET in Pc-frLHC.** The blue box shows the Chl *b* and carotenoid (Car) pool that is directly excited by the 460 nm laser. The orange, red, and dark-red boxes are the Chl pools of 680 nm–emitting bulk Chl, the 713 nm–emitting LWC$_{708}$ pool, and the 730 nm–emitting LWC$_{725}$ pool, respectively. The widths of the boxes roughly express the number of pigments contained in the pools. The blue and green arrows indicate the downhill and uphill energy transfers, respectively. The rate of the uphill energy transfer is temperature dependent. The winding blue arrow shows the energy migrations between the isoenergetic LWC$_{708}$ pigments. Red arrows indicate deexcitation by the fluorescence emission, dissipation to heat, etc. The blue dotted open arrow shows a possible energy exit for PSII. Approximate time constants are given beside the arrows.

LWC$_{725}$. The uphill EET from LWC$_{725}$ to bulk Chls was suggested from the data shown in Fig. 8a, and the action spectrum analysis of the cells of *P. crispa* demonstrated highly efficient PSII excitation with far-red light up to 750 nm[22]. Thus, the function of LWC$_{725}$ is probably to enhance light absorption in the far-red region.

Possible assignments of Chl *b* can be narrowed down based on the structure of Pc-frLHC and on the functional requirements of Chls. First, the Chls in LWCs and the exit site are likely Chl *a* due to their functional properties. Of the remaining three Chls (Chl601, Chl602, and Chl604), a possible candidate for Chl *b* is Chl604. Since Chl604 is isolated from the Chl network of Chl601-602-603-609/708-611-612-610-(613-614) in Pc-frLHC, Chl *b* at the position of 604 does not affect the entropy-driven uphill EET. If Chl *b* is a member of the cyclic Chl network, Chl *b* may hamper the entropy-driven uphill EET at an intermediary point such as that dividing the Chl *a* network.

The functional roles of the two carotenoids are difficult to discuss on the basis of the current experimental data. Pc-frLHC has only two carotenoids, violaxanthin and loroxanthin, at the central L1 and L2 sites, respectively. Lutein, loro, vio, β-carotene, and neoxanthin are known to exist in green algal LHCs[38]. Three functional roles of carotenoids in LHCs have been proposed: the transfer of absorbed light energy to Chls, the quenching of unwanted singlet and triplet excited states of Chls, and the stabilization of the protein structure[39–41]. Loro and vio in the Pc-frLHC can increase the cross section of absorption at around 500 nm. They can also accept excitation energy from triplet Chl or singlet oxygen and rapidly dissipate the energy by intersystem crossing. In recent report, red-shifted vio was observed in the LWC binding LHC of some Eustigma species that can use far-red light for PSII excitation, and red-shifted vio was reported to play a role as an energy acceptor from triplet red-shifted Chl *a* absorbing at 707 nm[42]. Therefore, the loro and vio in Pc-frLHC may function to dissipate the triplet excited state in the neighboring LWC candidates, Chl603-609-708-613′-614′, where the triplet excited Chls are formed with high probability.

In summary, Pc-frLHC with LWC$_{725}$ and LWC$_{708}$ can excite PSII by uphill EET using far-red light. The ratios of LWC$_{725}$, LWC$_{708}$, and bulk Chls in the Pc-frLHC undecamer are suitable for highly efficient uphill EET. Moreover, the ring-shaped arrangement of the 11 subunits enables the formation of an energetically connected Chl network with 11 excited-energy exit sites, increasing the probability of energy transfer

to PSII. We found pentameric Chls at the interfaces between subunits. These pentameric Chls probably function as LWC. Pc-frLHC is a well-designed molecular energy transfer machine using far-red light. Pc-frLHC contributes predominantly to the growth of *P. crispa* in the Antarctic terrestrial habitat through efficient photosynthesis with a far-red light.

## Methods

### Protein purification and characterization

Thylakoid membranes were prepared from thalli of *P. crispa* harvested from Antarctica as described in our recent report[22]. To purify Pc-frLHC, the thylakoids were diluted with Buffer-A (25 mM MES, pH6.5, 1 M betaine, 10 mM MgCl$_2$, 5 mM EDTA, 12.5% glycerol) and solubilized at 0.5 mg Chl ml$^{-1}$ with 1% dodecyl-β-D-maltoside (β-DDM) on ice for 20 min. The solubilized sample was centrifuged with an Optima™ TLX and TLA-110 rotor (Beckman Coulter, Brea, CA, USA) at 20,000 × *g* for 20 min. The supernatant was fractionated by sucrose density gradient (SDG) centrifugation at 267,000 × *g*, at 4 °C for 16 h. The middle green band containing PSI-LHCI and Pc-frLHC was collected and diluted 10 times with Buffer-B (25 mM MES, pH6.5, 1 M betaine, 0.03% β-DDM). The upper green band and the lowest light-green part were collected as LHCII and PSII-LHCII fractions, respectively. The diluted fraction was adsorbed on a diethylaminoethyl cellulose column (DE52; Whatman, Maidstone, UK), and the PSI-LHCI fraction was washed out with 150 mM NaCl containing Buffer-B. The Pc-frLHC fraction was eluted with 250 mM NaCl containing Buffer-B, diluted five times with Buffer-B, and precipitated by centrifugation at 417,000 × *g* at 4 °C for 2 h. For fluorescence measurements, pigment analysis, and cryo-EM analysis, the concentrated fraction was fractionated again by SDG centrifugation and the lowest dark-green band was collected. Sucrose was removed by dilution with Buffer-B and ultracentrifugation cycles.

The purified Pc-frLHC fraction was characterized by absorbance and fluorescence spectra measurements, high-resolution clear native (hrCN)-polyacrylamide gel electrophoresis (PAGE)[43], SDS-PAGE[44], and HPLC[45]. hrCN-PAGE was performed with stacking gel containing 2.91% acrylamide (A.A.) and 0.09% bis acrylamide (bis A.A.) and a separation gel comprising of linear gradient of 3.84-12.48% (w/v) of A.A., 0.12-0.39% (w/v) bis A.A. and 0-15.5% (w/v) glycerol. The gel was pre-run with anode buffer (25 mM imidazole/HCl, pH 7.0) and cathode buffer (0.03 % β-DDM, 50 mM Tricine, 7.5 mM imidazole, pH 7.0) with constant current of 4 mA at 4 °C for 30 min. LHCII, PSI, PSII-LHCII and Pc-frLHC fractions containing 0.03 % β-DDM were applied to the gel. The thylakoid sample was solubilized with 1% (w/v) β-DDM at a chlorophyll concentration of 0.5 mg Chl/ml for 20 min on ice and centrifugated at 20,000 × *g* at 4 °C for 10 min. The supernatant was applied to the gel. The electrophoresis was performed with constant voltage up to 100 V or 4 mA at 4 °C.

SDS-PAGE was performed with stacking gel containing 6% (w/v) A.A. and 0.08% (w/v) bis A.A. and separation gel comprising of linear gradients of 16-24% (w/v) A.A and 0.27−0.37% (w/v) bis A.A[44]. The gel contained 7.5 M urea, 0.375 M Tris-HCL, pH8.4 and 0.1% (w/v) SDS. Samples were dissolved in 2% (w/v) lithium dodecylsulfate, 60 mM dithiothreitol, 60 mM Tris-HCl, pH8.5 and 15% (w/v) sucrose at a chlorophyll concentration of c.a. 0.5 mg Chl/ml condition. The thylakoid sample was heated at 100 °C for 1 min to accelerate dissociation and denaturation of proteins. Following centrifugation of the treated sample at 20,000 × *g* for 10 min, the supernatants were applied to the gel. The electrophoresis was performed at room temperature with a constant current of 10 mA[44].

Absorption spectra were measured by MPS-2450 (Shimadzu, Kyoto, Japan). The slit width was set to 2.0 nm. The spectra were normalized by setting the absorbance at the peak of Qy band to 1. The absorbance at 750 nm were set to zero. Fluorescence emission spectra at room temperature were measured by a fluorescence spectrophotometer, RF-6000 (Shimadzu, Kyoto, Japan). A long pass filter that

passes light with wavelength longer than 350 nm and a filter that passes light with wavelength longer than 600 nm were placed in the excitation and the emission window, respectively, to eliminate higher-order light of monochromatic light. The slit widths of the excitation and emission sides were 5 nm and 3 nm, respectively. In the fitting analysis of the absorbance spectrum of purified Pc-frLHC, the peak wavelength of each component was estimated by the second and fourth derivatives of the absorbance spectrum, and fitting analysis with Gaussian functions was performed by Magic plot 2.7.2 (Magicplot Systems, St. Petersburg, Russia). Pigment analysis was performed using a HPLC system[45]. A μBondapak C18 column (100 × 8 mm, RCM-type; Waters, Milford, USA) and a guard column of the same material (Waters) were used for gradient analyses. The mobile phase was performed with a linear gradient from methanol:water (9:1, v/v) to methanol for 20 min and subsequent methanol elution at flow-rate of 1.8 ml/min. The purified Pc-frLHC fraction was injected directly into the HPLC system, and the absorbance at 440 nm was monitored.

### Amino acid sequence analysis

The Pc-frLHC protein was separated with SDS-PAGE, and the gel was stained with Coomassie brilliant blue (CBB) solution (0.1% CBB-R, 10% acetic acid, and 50% methanol) and destained with 25% methanol containing 10% acetic acid. The band at 29 kDa was cut out and put in a microtube for lysyl endopeptidase treatment. The peptidase treatment was performed according to a previously described protocol[46]. The fragmented peptides were separated by SDS-PAGE and electrophoretically blotted onto a PVDF membrane. The N-terminal amino acid sequences of four peptides were determined with a Procise 492cLC (Applied Biosystems, Carlsbad, CA, USA). The peptide sequence data were deposited in the UniProt Knowledgebase under accession number C0HLU5. We searched for the most probable cDNA of Pc-frLHC by using the determined internal peptide sequences from the total mRNA libraries of *P. crispa* (BioProject ID: PRJNA329112)[47]. The cDNA sequence of the Pc-frLHC gene was submitted to Third Party data (TPA) of the DDBJ/EMBL/GenBank databases and was assigned the accession number TPA: BR001753. The signal peptides and the transmembrane helices were predicted from the 3D structures registered in the PDB (Cr_Lhca2; 6JO5, 6IJO, Bc_LhcaJ; 6IGZ, Ds_Lhca5; 6SL5) and from the results of secondary structural prediction using Jpred 4[48] and TargetP-2.0[49]. Evolutionary analyses of LHCs were conducted with the neighbor-joining method in MEGA7[50].

### Cryo-EM sample preparation and data collection

For cryo-grid preparation, 3 μl of 6 mg protein ml⁻¹ Pc-frLHC (25 mM MES (pH 6.5), 0.015% β-DDM, and 0.5 M betain) was applied to a holey carbon grid (Quantifoil, Cu, R1.2/1.3, 300 mesh; Quantifoil, Jena, Germany), which was rendered hydrophilic by a 30 s glow-discharge in air (11 mA current) with a PIB-10 ion bombarder (Vacuum Device, Ibaraki, Japan). The grid was blotted for 5 s with a blot force of 25 at 18 ˚C and 100% humidity. The grid was then flash-frozen in liquid ethane using Vitrobot Mark IV (Thermo Fisher Scientific, Waltham, MA, USA). For automated data collection, 1555 micrographs were acquired with a Talos Arctica microscope (Thermo Fisher Scientific) operating at 200 kV in nanoprobe mode using EPU software. Micrograph movies were collected by a 4k × 4k Falcon 3EC direct electron detector (in electron counting mode) at a nominal magnification of 92,000 (1.13 Å/pixel). Fifty movie frames were recorded at an exposure of 1.00 e⁻/Å² frame, corresponding to a total exposure of 50 e⁻/Å². The defocus steps used were 1.0, 1.5, 2.0, 2.5, and 3.0 μm.

### Cryo-EM data processing

The cryo-EM data processing is summarized in Fig. 4 and Supplementary Fig. 4. First, the movie frames were aligned, dose-weighted, and averaged using an algorithm implemented on RELION3[51] on 5 × 5 tiled frames with a B-factor of 200. The nonweighted movie sums were used for Contrast Transfer Function (CTF) estimation with the Gctf program[52], while the dose-weighted sums were used for all subsequent steps of image processing. Particles were picked fully automatically using SPHIRE crYOLO[53,54] with a generalized model using a selection threshold of 0.02. RELION3 was used to perform the subsequent processes: 2D classification, ab initio reconstruction, 3D classification, 3D refinement, CTF refinement, and Bayesian polishing[51].

A stack of 696,095 particle images was extracted from the 1555 dose-weighted sum micrographs while rescaling to a box of 144 × 144 pixels in size at 3.39 Å/pixel and was subjected to 2D classification (200 expected classes). For ab initio map reconstruction, 195,767 particles corresponding to the best 11 classes, i.e., those that displayed secondary-structural elements and multiple views of Pc-frLHC, were selected. Two of those classes were clearly top views of a ring structure composed of 11 subunits with rotational symmetry (Supplementary Fig. 4c). Therefore, C11 symmetry was imposed on the generated ab initio map and used as an initial reference map for the 3D classification. For the subsequent 3D classification of four expected classes, 654,477 particles from 32 classes after 2D classification were selected with more relaxed criteria. The 3D volume and 185,680 particles of the 3D class with the highest resolution were used as the inputs of the subsequent 3D refinements with C11 symmetry. The refined volume and particle images were rescaled to a box of 432 × 432 pixels in size at 1.13 Å/pixel. The particle images that were duplicated as a result of alignments were excluded. Then, 160,157 selected particles were 3D auto-refined (C11 symmetry, with a mask diameter of 280 Å) twice (the first run was without and the second was with a soft-edged 3D mask). One cycle of Bayesian polishing and CTF refinement was performed, followed by 3D refinement with a soft-edged 3D mask (C11 symmetry, with a mask diameter of 360 Å) after each Bayesian polishing and CTF refinement step.

No-alignment 3D classification was then conducted (C11 symmetry, two expected classes, T = 4) with a soft-edged 3D mask, and 99,510 particles were selected by choosing the best 3D class. The last 3D refinement (C11 symmetry, with mask diameter 460 Å) with a soft-edged 3D mask generated the result at 3.13 Å resolution. The gold standard FSC resolution with a 0.143 criterion[55] was used as the global resolution estimation. The local resolution was estimated using an algorithm implemented on RELION3. UCSF Chimera[56] was used for visualization.

### Model building, refinement, and validation

The initial model was built using Map_to_Model[57] in PHENIX software[58]. The model was manually corrected by Coot[59], followed by Real-space Refinement[60] in PHENIX. The model was refined by multiple cycles of manual modifications in Coot and Real-space Refinement in PHENIX. NCS restraints were used for the automatic refinement. The refined model was validated using MolProbity[61] in PHENIX. FSC between the map and the model calculated by PHENIX[58] showed a resolution of 3.18 Å at FSC = 0.5. UCSF Chimera and PyMOL (Schrödinger, New York, NY, USA) were used for visualization.

### Fluorescence measurement

The time-resolved fluorescence spectra were measured by a streak scope (C1067; Hamamatsu Photonics, Hamamatsu, Japan). The magic angle configuration was used. For measurements with excitation at 740 nm, which selectively excites LWCs, a femtosecond Ti:S laser (MAITAI; Spectra-Physics, Santa Clara, CA, USA) was used as the excitation source. A short-pass filter with a cutoff wavelength of 700 nm (FESH0700; Thorlabs, Newton, NJ, USA) was set before the detector to block the excitation laser. For measurements with excitation at 460 nm, we used the second-harmonics pulse of the Ti:S laser generated by a barium borate (BBO) crystal and a long-pass colored glass filter with a cutoff of 480 nm. The instrumental response function was determined by measuring a standard sample (an aqueous solution of

Malachite Green) that was known to have a very short fluorescence lifetime[62]. For the steady-state fluorescence spectral measurements, a conventional fluorometer (F4500; Hitachi, Tokyo, Japan) was used.

The sample solution was set in the copper sample holder of a home-built Dewar. For the measurements at 273 K, 201 K, and 80 K, the Dewar was filled with ice in a liquid water bath, dry ice in a liquid ethanol bath, and liquid nitrogen as the cooling medium, respectively. The Pc-frLHC solution in the buffer was mixed with a two-fold volume of glycerol to maintain the transparency of the solution, except for the measurement of time-resolved fluorescence at 273 K.

### Estimation of the exciton coupling
We calculated the exciton coupling between molecules A and B using the following equation:

$$V_{A,B} = \frac{1}{\epsilon} \sum_{I=1}^{N} \sum_{J=1}^{N} \frac{q_I^A q_J^B}{r_{IJ}} \qquad (1)$$

Here, $q_I^{A/B}$ is the atomic transition charge of the $I$-th atom in molecule A/B and $r_{IJ}$ is the distance between the $I$-th atom in A and the $J$-th atom in B. $\epsilon$ (here set to 2.0) is the dielectric constant of the protein matrix which shields the electrostatic interaction. We used previously reported values of the atomic transition charges[63], which were obtained by quantum chemical calculations (Hartree–Fock and configuration interaction with single excitations) of the ground and excited states of Chl $a$ in a vacuum. The results of the calculations are shown in Supplementary Table 3.

### Analysis using the Arrhenius equation
The blue curves in Fig. 8b are the fitting curves to the sum of two Gaussian functions with peaks at 713 nm and 730 nm. The fitting was done under the constraint that the peak position of each band takes the same value for the data at every temperature. The inset in Fig. 8b shows the Arrhenius plot[64] of the ratio of the band areas of the 713 nm band with respect to that of the 730 nm band. The blue straight line is the fitting according to the equation

$$\frac{Area_{F713}}{Area_{F730}} = \frac{I_{F713}}{I_{F730}} \exp\left[-\frac{\Delta E}{k_B T}\right] \left( = \frac{N_{708}}{N_{725}} \exp\left(\frac{\Delta E}{k_B T}\right) = \frac{k_{725\rightarrow708}}{k_{708\rightarrow725}} \right) \qquad (2)$$

Here, $k_B$ is the Boltzmann constant. $I_{F713/F730}$ and $\Delta E$ are the intrinsic emission intensity of the Chl $a$ pool LWC$_{708/725}$ emitting the F713/F730 fluorescence band and the energy gap between the excited states of the two pigment pools, respectively. The intrinsic fluorescence intensity was that observed when the temperature would be infinite. In the present model, we assumed that the ratio of the intrinsic emission intensity is determined by the ratio of the pigment number ($N_{708/725}$) constituting the LWC$_{708/725}$. Then, the area ratio of the two fluorescence bands is equal to the ratio of the rates of the backward EET with respect to the forward one (the final term of Eq. 2). The fitting line is obtained with the value of $\Delta\Delta E$ fixed to 318 cm$^{-1}$ (= 458 K), which is calculated from the energy difference of the two peak wavelengths (Fig. 8b). Finally, we could estimate the value of $I_{F713}/I_{F730}$ as approximately 12, suggesting that the F713 emission is intrinsically about 12-fold brighter than the F730 one.

### Kinetic analysis of the EET
The fluorescence decay curves at various wavelengths were fitted to the sum of the exponential functions:

$$F_\lambda(t) = \left[ \sum_{i=1}^{n} A_i \exp\left(-\frac{t}{\tau_i}\right) H(t) \right] \otimes I(t) \qquad (3)$$

Here, $I(t)$ is the instrumental response function (Fig. 8a open circles), the symbol $\otimes$ indicates convolution, and $H(t)$ is the Heaviside step function. The fitting was done under the constraint that the time constants take the same values for every monitoring wavelength (global fitting). Three exponential components were found to be sufficient to fit the data. The pre-exponential factors plotted against the monitoring wavelength are called fluorescence decay–associated spectra (FDAS)[65] and are shown in Fig. 8c–e. The positive and negative signs in FDAS mean that the fluorescence at that wavelength has components that decay and rise, respectively, with the time constant of the FDAS. Thus, the energy transfer is reflected in the profile of FDAS having positive and negative signs on the shorter and longer wavelength sides, respectively. The above analysis was conducted using the software igor pro ver. 6 (WaveMetrics, Inc. Portland, USA).

### Reporting summary
Further information on research design is available in the Nature Portfolio Reporting Summary linked to this article.

## Data availability
The Cryo-EM map of Pc-frLHC is deposited in the Electron Microscopy Data Bank under accession code EMD-35080. Structural coordinates related to the cryo-EM map are deposited at the Protein Data Bank under accession code 8HW1. The peptide sequence data of Pc-frLHC were deposited in the UniProt Knowledgebase under accession number C0HLU5. The cDNA sequence of the Pc-frLHC gene was submitted to Third party data (TPA) of the DDBJ/EMBL/GenBank databases and was assigned the accession number TPA: BR001753. Source data are provided with this paper.

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

## Acknowledgements

We thank Ms. Yumiko Makino of the National Institute for Basic Biology (NIBB) for her support with the N-terminal amino acid sequence analysis. We also thank Dr. Hiroshi Imai of Osaka University and Dr. Keisuke Kawakami of RIKEN SPring-8 Center for their technical advice about the sample preparation for cryo-EM analysis. We thank Dr. Kotaro Koiwai of KEK for helping with 3D model building. This work was supported by Grants-in-Aid from the Japanese Ministry of Education, Culture, Sports, Science, and Technology (grant nos. 17K19431 to M.Ko., 19H03187 to Y.S.). This work was also supported by the Sumitomo Foundation (grant no. 151376 to M.Ko.), the National Institute of Polar Research through the General Collaboration Project (no. 31-29 to Y.Kas.), and the NIBB Collaborative Research Program (nos. 17-702, 18-506, and 19-704 to M.Ko). This work was partly supported by the Platform Project for Supporting Drug Discovery and Life Science Research (Basis for Supporting Innovative Drug Discovery and Life Science Research (BINDS)) from the Japan Agency for Medical Research and Development (AMED) under Grant No. JP20am0101071 and 22ama121001 to T.S. (supporting no. 1649). The cryo-EM data were collected at the cryo-EM facility in KEK (Ibaraki, Japan).

## Author contributions

M.Ko. conceived and designed the experiments. M.Ko. and S.K. collected the *Prasiola crispa* samples from Antarctica. M.Ko. purified the samples. M.Ko., S.T., and Y.Kas. performed biochemical analyses. M.Ko., Y.Kam., and K.H. performed primary sequence analysis. M.Ko., M.Ka., and N.A. prepared the cryo-EM samples and collected data. M.Ka. and T.M. processed the cryo-EM data. M.Ka. performed model building, refinement, and validation of the cryo-EM structure. M.Ko. and Y.S. performed the spectroscopic experiments and analysis. M.Ko., Y.S., and T.S. analyzed the data and interpreted the results with contributions from H.K., and wrote the manuscript.

## Competing interests

The authors declare no competing interests.
