## [Peer Review File · Nature Communications]

Uphill energy transfer mechanism for photosynthesis in an Antarctic algaReviewer #1 (Remarks to the Author):

This M/S really presents two things, the structure of the Pc-frLHC and spectroscopic work to try to explain its light harvesting properties. The structural work appears to have been well carried out even though the overall resolution is disappointing. On the other hand the spectroscopic side is far from convincing.

Specific points: The issue of the relevance of this work to astrobiology is extremely tenuous. In Fig. 1 c the gels are very poorly explained. Now you have the structure what is the true mass of the holo-complex? The absorption spectrum of the isolated complex versus the thylakoid membrane shows that the amount of the LHC is extremely small. Does it really have such a transformational significance? How did the authors decide which side of the complex was the stromal side and which side was the luminal side? In Fig4b where is the actual evidence for the proposed arrangement of the LHC with PS2?

It is stated that energy transfer from the long wavelength chlorophylls to PS2 is very efficient. This is not quantified anywhere, not even in the original paper in BBA describing this system. So how efficient is efficient?

The time resolved data presented in Fig2 is very hard to understand. It is not properly explained. For example no clear kinetic model of the proposed ET pathway is presented. What does excitation at 460 actually excite?

There is a similar vagueness about the assignment of those chlorophylls in the structure to the different absorption bands. Arguments are presented that might be true in some cases but the justification for their use in this specific case is weak.

Reviewer #2 (Remarks to the Author):

For several decades, photosynthesis remains the headliner topic for many biophysical, biochemical, genetical, and structural research. The contemporary advances in technological research methods elevated the study of photosynthesis to a new level, expanding the boundaries of understanding of hitherto little-known natural processes. Due to the wide range of adaptational abilities, photosynthetic organisms survive and prosper almost in any ecological niches of our Planet.

Particular interest arises in the organisms that can withstand extreme environmental conditions. The question of how the photosynthetic machinery adapts and efficiently functioning is currently the topic of study for many leading scientists all over the globe.

Expectedly the more primitive organisms possess broader adaptation variabilities (capacity, possibility). At the same time, the higher plants and algae, due to their native complexity, are more adaptational-restricted.

Nonetheless, in the current article, the group of scientists investigated the small terrestrial Antarctic green algae - *Prasiola crispa* that is exceptionally photosynthetically effective, living under extreme conditions including severe cold, drought, and intense sunlight.

The functioning features of the photosynthetic machinery that determines the successful survival strategy of the Antarctic algae are the subject of this article.

The most noteworthy results of this article are:

a) The presence of circular (bacterio-like) light-harvesting complexes (LHCII), named Pc-frLHC.

b) The unique features of the PC-frLHC antenna that appears to be the key adaptation factor for the survival of *Prasiola crispa* are the following:

- Undecameric 11-fold symmetry;

- Exclusive subunit's structure with unprecedented Chls arrangement capable of long- wavelength absorbance. Structurally it is manifested in the presence of 2 low-wavelength Chls' - LWC708 & LWC725. Such structure leads to 2-steps uphill EET needed for PSII activation with far-red light.

c) The origin of Pc-frLHC refers to the PSI-LHCI family rather than PSII-LHC

II.

In this article, a combination of methods and techniques were utilised to analyse and characterise the structure and energy transfer machinery of Pc-FrLHC, including biochemical, spectroscopic, chromatographic and structural. The revealed results are cross-validated by well-selected research techniques.

Meanwhile, I have few major comments:

1. Non-investigated central cavity (hole) inside the Pc-frLHC antenna.

The space in the centre of the antenna ring most definitely is filled in with some lipids that potentially can influence the structural assembly. Apart from the fact that: "PSII is too large to fit into the inner space of the ring", very little is discussed about it.

- The lipidomic study can be suitable to investigate the central cavity containment.
- From the image processing side – the authors may try to
 - a) Run the 3D refinement with C1 symmetry.
 - b) Mask all but the central cavity (hole) and run the 3D classification.

Above mentioned steps may help the analysis and add the description of the central cavity (hole) structure of the Pc-frLHC antenna.

2. Pigments study

2.1 Chl b identification.

Although thorough biochemical and spectroscopic analyses identified the Chl b, the cryo-EM study did not position it within the Pc-frLHC antenna.

The question remains about the detailed energy transfer within the Pc-frLHC antenna and the role of Chl b in it.

- This question may be solved by resolution improvement and/or testing and changing the detergent during the isolation of the Pc-frLHC antenna with the following cryo-EM & image processing analysis.

2.2 Impact of the carotenoids on the energy transfer within the Pc-frLHC antenna.

Although both Chl and carotenoid were found, the main focus of this research lies in the study and the analysis of Chls. It is worth mentioning that the authors identified the presence of 2 types of carotenoids, which are shown in Figure 3 and Extended Data Figure 8.

The open question remains about the carotenoid's involvement in the energy transfer process.

- Adding more discussion about the carotenoid's impact can provide the required clarifications and shed more light on the EET mechanism.

As an open question –

Since the Pc-frLHC antenna is phylogenetically closer to the LHCI antenna, does the mechanism of excitation energy transfer from the Pc-frLHC antenna to PSI-LHCI occur?

Perhaps it may be the topic for future studies.

The other corrections, propositions, suggestions:

7 ...photosynthetic active radiation... photosynthetically active radiation.

56 in *P. crispa*, Pc-frLHC... The abbreviation Pc-frLHC is given for the first time.
Please provide a full name.

73 ...based light harvesting complexes (LHCs).
Please add the reference.

105 ...far-red light harvesting system requires...

Please change to light-harvesting since "light-harvesting" is majorly used.

108 ...EET with visible light.

Please add the reference.

122 While the exact molecular mass of Pc-frLHC was difficult to determine...

Why? Was the Pc-frLHC not stable? Did you try different detergents?

178-179 While inter-subunit protein-protein interactions are tight at the stromal side, there are only limited intersubunit interactions at the luminal side.

How could you explain that?

179 Due to the asymmetric interactions between the subunits, there is a large cavity at the interface of the two subunits How certain are you about this statement? Do you know what is located in the centre of the cavity? If not, please rephrase.

180-181 The cavity accommodates nine Chl molecules (Fig. 4). What do you mean by that? Are those Chl located inside the cavity? In "Figure 4 The chlorophyll network in Pc-frLHC", it is shown that the Chl is located around the cavity. Please clarify.

255 Thus, eleven PSII binding sites...

Not clear, I would add the Pc-frLHC, e.g. Thus, eleven PSII binding sites of Pc-frLHC.

254-255 Thus, eleven PSII binding sites can play an equivalent role in the energy transfer to PSII.

It is not clear how and where the Pc-frLHC complex is adjacent to PSII. Do you mean that the connection of Pc-frLHC to PSII is equally possible from any of the binding sites of Pc-frLHC?

259-260 When five acceptor complexes are bound, P_{exit} reaches 95%

Please check the front – it is different from the main text front.

401 Pc-frLHC absorbs far red light...

Please check the consistency. Change to far-red.

862 Pc-frLHC absorbs far red light...

Please check and change to far-red (as it was "far-red" in the text above).

844 Figure 3

In Figure 3, we see the model built based on the cryo-EM structure. It is possible or a) Specify the title or description; b) add the cryo-EM map in Figure 3a with high opacity (almost transparent).

844-845 Figure 3. a: Top (upper panel) and side

views (lower panel) are shown...

Please add the description of which side the top view is (luminal/stromal?).

847 Figure 3. c: Chlorophyll a arrangement... Subsequently, some subunits are coloured into a

similar colour: on the 3 o'clock – green-blue; on the 6 o'clock – almost the same green-blue. You may consider changing the colour in one of the cases (I guess 3 o'clock is the easiest way).

855 Figure 4 The chlorophyll network...a: Connections among...

Please add the view definition, e.g., top view, or parallel to the membrane plane.

968 Extended Data Figure 6 c: Magnification of two top views

An exciting indication of the structured matter presence in the hole. 2D class average image with "*" seems to contain heterogeneous particles. Did you have some broken particles?

1000 Extended Data Table 1: 604 none

Was the ligand not found for this Chl?

In conclusion, the methodology and the practical experiments described in this article were planned and performed to high standards and consistent with the aims and objectives of the study.

I consider this article of high impact and ready for publication in Nature Communications journal after minor corrections.

Sincerely

Point-to-point responses to the reviewers

#NCOMMS-21-29394-T “Uphill energy transfer mechanism for photosynthesis in the Antarctic alga”

by Makiko Kosugi et al.

Thank you very much for providing important comments. We are thankful for the time and energy the reviewers expended. Our responses to the reviewers' comments are as follow:

Comments from the reviewers are in italics.

Reviewer #1

Comment 1: *This M/S really presents two things, the structure of the Pc-frLHC and spectroscopic work to try to explain its light harvesting properties. The structural work appears to have been well carried out even though the overall resolution is disappointing. On the other hand the spectroscopic side is far from convincing.*

Response: Although the resolution of Pc-frLHC is 3.13Å, we consider that this is sufficient to discuss the structural basis of EET. The orientations of Chl *a* molecules are clearly determined using the current cryo-EM map, and these data could be used for the theoretical calculations of exciton coupling.

Our spectroscopic data clearly revealed the existence of two distinct LWCs, which absorb far red light and play a role in EET. This is the first report to provide the structural basis for the far-red light utilization by the shaded algal PSII. For further details, spectroscopic data should be assigned. However, it is still a very difficult task to theoretically assign the excitation energy of each bound Chl for the most advanced quantum chemistry. A combined analysis using mutagenesis, spectroscopy, and structural analysis may be another approach to the above task. These analyses are clearly beyond the scope of this manuscript and should be performed as future projects.

Comment 2: *Specific points: The issue of the relevance of this work to astrobiology is extremely tenuous.*

Response: Thank you for pointing this out. We have deleted the description about astrobiology from the main text.

Comment 3: *In Fig. 1 c the gels are very poorly explained. Now you have the structure what is the true mass of the holo-complex?*

Response: Thank you for pointing this out. To address this, we added a description of the molecular mass based on the Pc-frLHC structure. In addition, since CN-PAGE is not an appropriate modality to determine the molecular mass, we revised the relevant sentences.

Comment 4: *The absorption spectrum of the isolated complex versus the thylakoid membrane shows that the amount of the LHC is extremely small. Does it really have such a transformational significance?*

Response: Yes, we consider it is significant. *Prasiola crispa* has two major LHCs that can deliver energy to PSII, LHCII and Pc-frLHC. Of the two LHCs, only Pc-frLHC can absorb the far-red light (page 3, lines 108-112). Extended Data Fig. 1a shows that little visible light reaches the lower layer of the colonies; only the far-red light can be utilized in the lower layer of the colonies. While *P. crispa* in the lower colonies expressed a large amount of LHCII for use in environments with limited visible light, LHCII cannot be excited by the far-red light. Indeed, *P. crispa* without Pc-frLHC cannot grow in far-red light alone (Kosugi, unpublished). Thus, we believe that LHCII barely contributes to the survival of *P. crispa* in lower colonies.

In contrast, since the Pc-frLHC can utilize the far-red light, we consider that the Pc-frLHC is essential for *P. crispa* propagation in the lower layer colonies. The analysis of the action spectrum and the absorption spectrum of thylakoids revealed that the global quantum yield of the PSII activity by far-red light was nearly the same as that by visible light (Kosugi 2020). Furthermore, our preliminary results showed that the relative amount of Pc-frLHC in the lower layer is larger than that of the surface colonies. According to experiments using cultured *P. crispa*, Pc-frLHC is not expressed under normal light conditions but seems to be induced under weak or long wavelength light conditions. These results are consistent with the fact that *P. crispa* in the lower part of the colonies showed higher absorbance of the far-red light than that observed on the surface (Extended Data Fig. 1c-e). Taken together, our findings led us to conclude that Pc-frLHC in thylakoid membranes is likely to play a significant role in the photosynthesis and growth of *P. crispa* in the natural habitat.

Comment 5: *How did the authors decide which side of the complex was the stromal side and which side was the luminal side?*

Response: The stromal and luminal sides of Pc-frLHC were determined from accumulated information of other LHCs. All LHCs of algae and plants have evolved from a common ancestral LHC and have the same basic structure of three transmembrane helices (-A,-B and -C). No exceptions have been found yet. Kühlbrandt et al. determined the first tertiary structure of LHCII and assigned the N-terminus facing the stromal side (Kühlbrandt et al. 1994). After this structure, the tertiary structures of the LHCII and PSII super complexes and the LHCI and PSI super complexes were determined (Ben-Shem et al. 2003, Wei et al. 2016). All the LHCs determined are aligned in the same manner. Based on these data, we assigned the stromal and luminal sides. We have added the description in the result section (page 4, lines 141-143).

Comment 6: *In Fig4b where is the actual evidence for the proposed arrangement of the LHC with PS2?*

Response: The PSII–Pc-frLHC arrangement in Fig. 4b in the original manuscript (see below) is just a model based on tentative assignments of Chls. While we have described the possible assignments of Chl in the revised manuscript, we removed the definitive descriptions of the assignments.

Fig. 4 in the original manuscript

Comment 7: *It is stated that energy transfer from the long wavelength chlorophylls to PS2 is very efficient.*

This is not quantified anywhere, not even in the original paper in BBA describing this system. So how efficient is efficient?

Response: Thank you for the comments. We regret that our original description of the energy transfer efficiency was not clear. We revised the description for clarity (page 3, lines 96-98). The analysis of the action spectrum and the absorption spectrum revealed that the global quantum yield of the PSII activity by far-red light was nearly the same as that by visible light (Kosugi 2020). Furthermore, PSII was slightly excited by the far-red light in cultured cells that do not express Pc-frLHC (Kosugi 2020). This result is consistent with the fact that Pc-frLHC was the only LHC that can absorb far red light (page 3, lines 108-112).

To determine the exact quantum yield of Pc-frLHC, purification of the super complex of PSII and Pc-frLHC is needed. However, purification of the intact Pc-frLHC-PSII super complex will be a future research project.

Comment 8: *The time resolved data presented in Fig2 is very hard to understand. It is not properly explained. For example no clear kinetic model of the proposed ET pathway is presented.*

What does excitation at 460 actually excite?

Response: Thank you for the comment. To clarify the kinetic model of Pc-frLHC, we have added Fig. 5, which shows the kinetic parameters obtained in this study.

The light at 460 nm excites Chl *b* and carotenoids. Our analysis revealed that the light energies absorbed by Chl *b* and carotenoids are transferred to LWC through the bulk Chl *a*. On the other hand, the 740 nm light was used for the selective excitation of the LWCs.

Comment 9: *The is a similar vagueness about the assignment of those chlorophylls in the structure to the different absorption bands. Arguments are presented that might be true in some cases but the justification for their use in this specific case is weak.*

Response: Thank you for the valuable comments. As pointed out, the assignment of Chls in the Pc-frLHC complex is not definitive but merely tentative. However, the assignment based on the structure and calculated coupling constants seems to be reasonable and meets the standard of this field. Ultimately, the assignment should be confirmed using a site-directed mutagenesis study. However, such an analysis is beyond the scope of this study. Cloning and establishment of an expression system of the active protein will be carried out as another project.

Reviewer #2

Comment 1: *For several decades, photosynthesis remains the headliner topic for many biophysical, biochemical, genetical, and structural research. The contemporary advances in technological research methods elevated the study of photosynthesis to a new level, expanding the boundaries of understanding of hitherto little-known natural processes. Due to the wide range of adaptational abilities, photosynthetic organisms survive and prosper almost in any ecological niches of our Planet.*

Particular interest arises in the organisms that can withstand extreme environmental conditions. The question of how the photosynthetic machinery adopts and efficiently functioning is currently the topic of study for many leading scientists all over the globe.

Expectedly the more primitive organisms possess broader adaptation variabilities (capacity, possibility). At the same time, the higher plants and algae, due to their native complexity, are more adaptational-restricted.

*Nonetheless, in the current article, the group of scientists investigated the small terrestrial Antarctic green algae - *Prasiola crispa* that is exceptionally photosynthetically effective, living under extreme conditions including severe cold, drought, and intense sunlight.*

The functioning features of the photosynthetic machinery that determines the successful survivance strategy of the Antarctic algae are the subject of this article.

The most noteworthy results of this article are:

*a) The presence of circular (bacterio-like) light-harvesting complexes (LHCII), named Pc-frLHC.
b) The unique features of the PC-frLHC antenna that appears to be the key adaptation factor for the survivance of *Prasiola crispa* are the following:*

- Undecameric 11-fold symmetry;

- Exclusive subunit's structure with unprecedented Chls arrangement capable of long-wavelength absorbance. Structurally it is manifested in the presence of 2 low-wavelength Chls' - LWC708 & LWC725. Such structure leads to 2-steps uphill EET needed for PSII activation with far-red light.

c) The origin of Pc-frLHC refers to the PSI-LHCI family rather than PSII-LHCII.

Response: Thank you for sharing our enthusiasm for the structural study of Pc-frLHC. Our study provides an important advance in the uphill energy transfer that is required for far-red photosynthesis.

Comment 2: *In this article, a combination of methods and techniques were utilised to analyse and characterise the structure and energy transfer machinery of Pc-FrLHC, including biochemical, spectroscopic, chromatographic and structural. The revealed results are cross-validated by well-selected research techniques.*

Response: Thank you for the positive comments.

Comment 3: *Meanwhile, I have few major comments:*

1. Non-investigated central cavity (hole) inside the Pc-frLHC antenna.

The space in the centre of the antenna ring most definitely is filled in with some lipids that potentially can influence the structural assembly. Apart from the fact that: “ PSII is too large to fit into the inner space of the ring”, very little is discussed about it.

- *The lipidomic study can be suitable to investigate the central cavity containment.*
- *From the image processing side – the authors may try to*

a) Run the 3D refinement with C1 symmetry.

b) Mask all but the central cavity (hole) and run the 3D classification.

Above mentioned steps may help the analysis and add the description of the central cavity (hole) structure of the Pc-frLHC antenna.

Response: Thank you for the valuable comments. We consider that the unidentified cryo-EM density in the hole of the Pc-frLHC complex would be detergents based on some experimental results. First, we used detergent (dodecyl- β -D-maltoside) to exchange lipids associated with Pc-frLHC. Then, the solubilized Pc-frLHC by the detergent was used for the cryo-EM analysis. Second, the Pc-frLHC sample used for the cryo-EM analysis is very pure, as indicated by SDS-PAGE (**Fig. 1c**). Therefore, protein contaminations are also unlikely. Third, we tried the 3D refinement with C1 symmetry (resolution 4.1 Å). In addition, since the 3D classification with C1 symmetry clearly separated the central cavity (hole) with/without visible density, we further employed the “partial signal subtraction” approach implemented in Relion and obtained a map with 11.9 Å resolution. While we could observe a cryo-EM density in the hole of the ring structure, the cryo-EM density could not be interpreted. For example, the subunit structure of Pc-frLHC could not explain the density. It is likely that pseudo correlations caused by the 3D refinement of Relion resulted in the cryo-EM density in the hole. Fourth, since the inner surface of the hole is lined by hydrophobic amino acid residues, it is reasonable to assume that these residues interact with molecules that can interact with hydrophobic residues. Based on these results, it seems reasonable to predict that the cryo-EM density inside the hole represents the detergent used in the purification, although we can not exclude the possibility that the remaining lipid in the hole caused the cryo-EM density.

We have added some sentences discussing the cryo-EM density inside the hole in the Result section (page 4, lines 148-156).

Comment 4: *2. Pigments study*

2.1 Chl b identification.

Although thorough biochemical and spectroscopic analyses identified the Chl b, the cryo-EM study did not position it within the Pc-frLHC antenna.

The question remains about the detailed energy transfer within the Pc-frLHC antenna and the role of Chl b in it.

• This question may be solved by resolution improvement and/or testing and changing the detergent during the isolation of the Pc-frLHC antenna with the following cryo-EM & image processing analysis.

Response: Thank you for the comment. First, it is very hard to distinguish between Chl *a* and Chl *b* using the current cryo-EM map, because the structural differences between the two Chls are very subtle. To distinguish the two Chls, a very high-resolution cryo-EM map (very close to atomic resolution) is needed. Unfortunately, such maps are still extremely rare, and much larger numbers of images (and amounts of samples) than used in the current study would be required to achieve this level of high-resolution with a membrane protein using cryo-EM. However, we could analyze the energy transfer from Chl *b* in Pc-frLHC (**Extended Data Fig. 9**). The analysis revealed that the energy absorbed by Chl *b* and carotenoids is transferred to the bulk Chls and then moves to LWC (**Extended Data Fig. 9**).

While the reviewer suggested a good experiment plan, we would like to perform this as another project. The amount of Pc-frLHC sample available to us is currently very limited, since it is naturally sourced from Antarctica.

Comment 5: 2.2 Impact of the carotenoids on the energy transfer within the Pc-frLHC antenna. Although both Chl and carotenoid were found, the main focus of this research lies in the study and the analysis of Chls. It is worth mentioning that the authors identified the presence of 2 types of carotenoids, which are shown in Figure 3 and Extended Data Figure 8.

The open question remains about the carotenoid's involvement in the energy transfer process.

• Adding more discussion about the carotenoid's impact can provide the required clarifications and shed more light on the EET mechanism.

Response: Thank you for the important comment. As you recommend, we added an analysis of energy transfer from Chl *b* and carotenoids (**Extended Data Fig. 9**). We also added a discussion of a possible functional role of the carotenoids in the revised manuscript (page 10, lines 351- page 11, lines 368).

Extended Data Figure 9 | Comparison of the

excitation spectrum monitored at 730 nm to the 1-transmittance spectrum. Both spectra were measured at 77 K.

Comment 6: *As an open question –*

Since the Pc-frLHC antenna is phylogenetically closer to the LHCI antenna, does the mechanism of excitation energy transfer from the Pc-frLHC antenna to PSI-LHCI occur? Perhaps it may be the topic for future studies.

Response: Thank you for the valuable comment. As the reviewer suggested, it would be possible to transfer the energy from Pc-frLHC to PSI-LHCI. If Pc-frLHC plays a role in distributing far-red light energy between PSII and PSI, elucidation of the regulation mechanism of the energy distribution would be an interesting research topic.

Comment 7: *The other corrections, propositions, suggestions:*

7 ...photosynthetic active radiation...  photosynthetically active radiation.

Response: We corrected this phrase as the reviewer indicated.

Comment 8: *56 in P. crista, Pc-frLHC... The abbreviation Pc-frLHC is given for the first time.*

Please provide a full name.

Response: We have added the full name.

Comment 9: *73 ...based light harvesting complexes (LHCs).*

Please add the reference.

Response: We have added a reference.

Comment 10: *105 ...far-red light harvesting system requires...*

Please change to light-harvesting since "light-harvesting" is majorly used.

Response: We changed the word as the reviewer suggested.

Comment 11: *108 ...EET with visible light.*

Please add the reference.

Response: We have added a reference.

Comment 12: *122 While the exact molecular mass of Pc-frLHC was difficult to determine...*

Why? Was the Pc-frLHC not stable? Did you try different detergents?

Response: Thank you for the comments. This sentence was unclear and misleading. In our revision of the manuscript, we clarified this point about the molecular mass of Pc-frLHC (page3 line113-115). Please see also **Response 3 to Reviewer 1**.

Comment 13: *178-179 While inter-subunit protein-protein interactions are tight at the stromal side, there are only limited intersubunit interactions at the luminal side.*

*How could you explain that?*

Response: Thank you for the comments. Due to the limited inter-subunit interactions at the luminal side, it is possible to have a relatively large inter-subunit space for accommodating eight Chl molecules.

Comment 14: 179 *Due to the asymmetric interactions between the subunits, there is a large cavity at the interface of the two subunits How certain are you about this statement? Do you know what is located in the centre of the cavity? If not, please rephrase.*

Response: Thank you for the comment. Following the suggestion of the reviewer, we have rephrased the sentence in question.

Comment 15: 180-181 *The cavity accommodates nine Chl molecules (Fig. 4). What do you mean by that? Are those Chl located inside the cavity? In “Figure 4 The chlorophyll network in Pc-frLHC”, it is shown that the Chl is located around the cavity. Please clarify.*

Response: We apologize for the confusing description. The cavity means the space between subunits accommodating the eight Chls (Chl601, 603, 609, 708, 611', 612', 613', 614'). To avoid misunderstanding, we rewrote the sentences in question and added some additional description (page 4 line 146-148).

Comment 16: 255 *Thus, eleven PSII binding sites...*

*Not clear, I would add the Pc-frLHC, e.g. Thus, eleven PSII binding sites of Pc-frLHC.*

Response: We modified the sentences as the reviewer suggested (page 9 line 327).

Comment 17: 254-255 *Thus, eleven PSII binding sites can play an equivalent role in the energy transfer to PSII.*

*It is not clear how and where the Pc-frLHC complex is adjacent to PSII. Do you mean that the connection of Pc-frLHC to PSII is equally possible from any of the binding sites of Pc-frLHC?*

Response: We modified the sentences as the reviewer suggested (page 9 line 326-327).

Comment 18: 259-260 *When five acceptor complexes are bound, P_{exit} reaches 95%*

*Please check the font – it is different from the main text font.*

Response: Thank you for the comments. We corrected the font.

Comment 19: 401 *Pc-frLHC absorbs far red light...*

*Please check the consistency. Change to far-red.*

Response: We corrected as the reviewer suggested.

Comment 20: 862 *Pc-frLHC absorbs far red light...*

Please check and change to far-red (as it was “far-red” in the text above).

Response: Thank you for the close reading. We standardized the usage of this phrase as suggested.

Comment 21: 844 Figure 3

In Figure 3, we see the model built based on the cryo-EM structure. It is possible or a) Specify the title or description; b) add the cryo-EM map in Figure 3a with high opacity (almost transparent).

Response: We have modified the figure as the reviewer suggested.

Comment 22: 844-845 Figure 3. a: Top (upper panel) and side views (lower panel) are shown...

Please add the description of which side the top view is (luminal/stromal?).

Response: Thank you for the comments. We have revised the legend of **Fig. 3** (**Fig. 2** in the revised manuscript).

Comment 23: 847 Figure 3. c: Chlorophyll a arrangement... Subsequently, some subunits are coloured into a similar colour: on the 3 o'clock – green-blue; on the 6 o'clock – almost the same green-blue. You may consider changing the colour in one of the cases (I guess 3 o'clock is the easiest way).

Response: Thank you for the comment. We have changed the color of the subunit following the suggestion of the reviewer.

Comment 24: 855 Figure 4 The chlorophyll network...a: Connections among...

Please add the view definition, e.g., top view, or parallel to the membrane plane.

Response: Thank you for the comments. We have revised the legend of **Fig. 4** (**Fig. 3** in the revised manuscript).

Comment 25: 968 Extended Data Figure 6 c: Magnification of two top views

An exciting indication of the structured matter presence in the hole. 2D class average image with “*” seems to contain heterogeneous particles. Did you have some broken particles?

Response: As we described above (Response 3 to Reviewer 2), it was difficult to visualize the matter inside the hole, even when analyzing with C1 symmetry. We observed only a few broken particles in some micrographs. No other significant signs of the denaturation were observed.

Comment 26: 1000 Extended Data Table 1: 604 none

Was the ligand not found for this Chl?

Response: Thank you for the comments. Since there are no possible residues that can coordinate the Mg ion in the Chl604, we expect that the ligand is a water molecule. We have added a comment in

the footnote of the Extended Data Table 1. Identification of the water ligand is difficult without a resolution higher than 2.0 Å.

Comment 27: *In conclusion, the methodology and the practical experiments described in this article were planned and performed to high standards and consistent with the aims and objectives of the study.*

I consider this article of high impact and ready for publication in Nature Communications journal after minor corrections.

Response: Thank you for the positive comments. We appreciate the reviewer's valuable and thoughtful comments.

Reviewer #2 (Remarks to the Author):

The authors conducted tedious work to improve the manuscript by addressing my questions. I can recommend the manuscript for publication in the Nature Communications journal.

Kind regards,
Dmitry Semchonok

Reviewer #3 (Remarks to the Author):

The manuscript by Kosugi et al. describes a ring-shaped LHC undecamer from *Prasiola crispa*, a green alga in Antarctica. The authors solved the cryo-EM structure at 3.13 angstrom resolution and performed spectroscopic analyses of this Pc-frLHC supercomplex. Pc-frLHC represents the first example of a eukaryotic LHC that forms a ring-shaped structure. Pc-frLHC can absorb far-red light and excite PSII with high efficiency. However, Pc-frLHC is different from other LHCIIs in possessing four TMHs, but is similar to *Chlamydomonas* Lhca2. The authors suggested that the red chlorophylls in Pc-frLHC are the Chl 603-609 pair together with Chl 708, the latter is not present in other LHCs. Overall, this work is an interesting addition to the field of photosynthesis research. I only have a few suggestions/comments for the authors to consider when revising.

- Both *Prasiola crispa* and *Chlamydomonas reinhardtii* belong to green algae and both contain a specific LHC protein with four TMHs. CrLhca2 is the peripheral antenna of PSI, while Pc-frLHC was shown to transfer excitation energy to the PSII core. Therefore, it is very interesting to know the similarities and differences between the two types of LHCs in terms of protein structure and pigment arrangement. I encourage the authors to provide a supplementary figure showing the superimposed result of the two LHCs. A high-resolution structure of *Chlamydomonas* PSI-LHCII (7DZ7) was reported last year. The structural model of Lhca2 in this structure is almost complete, and can be used for comparison.

- In plants and other green algal PSI-LHCI complexes, four normal LHCI (with three TMHs) form a half moon-shaped belt that attaches to one side of the PSI core. It seems that three such LHCI belts or eleven normal LHCI might also be able to form a circle. I assume that *Prasiola crispa* also contains normal LHCI, so I would like to hear the author's comments on what is the significance of this green alga adopting Pc-frLHC (with four TMHs) to form a ring structure? Could the authors compare the structures of the Pc-frLHC undecamer and the normal LHCI belt to see what is the difference between them, and whether the fourth TMH of the Pc-frLHC contributes to these differences.

- Line 286, I don't think that Glu is the ligand for Chl 606 in other LHCI, it does not axially ligate the central Mg atom of Chl 606 in these structures. According to the structure of spinach LHCII and others, Chl 606 may be coordinated by a water molecule.

Reviewer #4 (Remarks to the Author):

The findings reported in this article are stunning, but they are not (yet) sold as such. The authors could consider hiring a scientific writer. Overall it should be explained more why this data shows an important step forward in the field of photosynthesis.

The abstract could be improved a lot based on the summary of the results provided by Reviewer 2: "The most noteworthy results of this article are:

a) The presence of circular (bacterio-like) light-harvesting complexes (LHCII), named Pc-frLHC.

b) The unique features of the PC-frLHC antenna that appears to be the key adaptation factor for the survival of *Prasiola crispa* are the following:

- Undecameric 11-fold symmetry;

- Exclusive subunit's structure with unprecedented Chls arrangement capable of long- wavelength

absorbance. Structurally it is manifested in the presence of 2 low-wavelength Chls' - LWC708 & LWC725. Such structure leads to 2-steps uphill EET needed for PSII activation with far-red light.
c) The origin of Pc-frLHC refers to the PSI-LHCI family rather than PSII-LHCII."

I have some issues with the way the data is described and/or interpreted:

Line 231-233 "Since only Chl b and the two carotenoids are excited in this analysis, this result suggests that the absorbed energies of Chl b and the two carotenoids are exclusively transferred to the 730 nm emitting species through Chl a." Please specify in which region only Chl b and Cars are excited. For the 400 nm to 550 nm region this statement is obviously not correct.

Line 245-247: "Importantly, the above model requires that LWC708 intrinsically emits stronger fluorescence than LWC725." What is the definition of intrinsic fluorescence? From the M and M section I understand that you mean the emission intensity if the energy would be the same or if the temperature would be infinite. This is not very clear. However the outcome is still valid. Your analyses suggests that the pigment pool contributing to the F713 emission is 12 times larger than that contributing to the F730 nm band. This is only valid when the radiative rates of both pools is equal. Strongly coupled pigments are known to be super-radiative, so this should be considered.

Line 334-336: The three factors described here are hardly related. The statements "problems remain" sound very negative. However, the problems described are not so large. The function of carotenoids are most likely the same as in every LHC (light absorption, structure stabilisation, photoprotection). Unsuccessful assignment of Chl b is a pity, but no serious issue. The function of LWC725 is probably just to enhance the light absorption in the far-red region.

Line 339-342: Point 1: Do you mean high probability to be excited after energy-transfer? The initial excitation probability will be low. Point 2: I don't understand why to authors speculate about a role in quenching why their data shows that uphill energy transfer is fast (Fig 4a).

Line 358-359: "In addition, loro seems to energetically connect an isolated Chl604 and the Chl a network at the stromal side." Energy transfer from the Chl604 through loro to the other Chls seems highly unlikely. I believe Chl to Car energy transfer has only been reported in the context of quenching.

Line 1001, Fig 5: The Chl b + Car to Bulk Chl arrow should be broader to show very fast energy transfer. The blue arrow in the LWC708 band is not described in the text. The dashed arrow should be broader than the ~25 ps arrow as it would otherwise not be very competitive with the transfer to the LWC708, at least when this describes the transfer at physiological relevant conditions (this info should be added as some green uphill-arrows are shown, of which the rate is of course temperature dependent). The emission of the LWC725 band also occurs on the ns time-scale, this could be added.

Minor issues:

Line 50-52 "Cells of deeper layer escape from photodamage at the sacrifice of photosynthetically active radiation except infrared." The sentence is hard to follow and infrared light is usually not considered as photosynthetically active radiation for green algae.

Line 52-53 "At deeper layers, *P. crista* achieves effective photosynthesis by low energy far-red light for photosystem II (PSII) excitation." by being able to use far-red ...

Line 73-74: The LWC in LHCI are only 2 Chls, I don't think that classifies as an aggregate of molecules.

Line 81-82: "The EET mechanism of the far-red light-capturing system is noteworthy from the perspective of energy yield." What is meant with this sentence?

Line 1034: "Shorter wavelengths than 400 nm of the excitation light were cut by long-pass filters."

Was this light present due to the present of the second harmonic?

Point-by-Point Responses

Comments from the reviewers are shown in italics. All page and line numbers are for the revised manuscript (marked manuscript).

REVIEWER COMMENTS

Reviewer #2

R2-01) *The authors conducted tedious work to improve the manuscript by addressing my questions. I can recommend the manuscript for publication in the Nature Communications journal.*

Response to R2-01) Thank you for the positive comment. We have further improved the manuscript following suggestions from reviewers 3 and 4.

Reviewer #3

R3-01) *The manuscript by Kosugi et al. describes a ring-shaped LHC undecamer from Prasiola crispa, a green alga in Antarctica. The authors solved the cryo-EM structure at 3.13 angstrom resolution and performed spectroscopic analyses of this Pc-frLHC supercomplex. Pc-frLHC represents the first example of a eukaryotic LHC that forms a ring-shaped structure. Pc-frLHC can absorb far-red light and excite PSII with high efficiency. However, Pc-frLHC is different from other LHCII in possessing four TMHs, but is similar to Chlamydomonas Lhca2. The authors suggested that the red chlorophylls in Pc-frLHC are the Chl 603-609 pair together with Chl 708, the latter is not present in other LHCs. Overall, this work is an interesting addition to the field of photosynthesis research. I only have a few suggestions/comments for the authors to consider when revising.*

Response to R3-01) Thank you for sharing our enthusiasm for our work. We have revised the manuscript following your comments.

R3-02) *Both Prasiola crispa and Chlamydomonas reinhardtii belong to green algae and both contain a specific LHC protein with four TMHs. CrLhca2 is the peripheral antenna of PSI, while Pc-frLHC was shown to transfer excitation energy to the PSII core. Therefore, it is very interesting to know the similarities and differences between the two types of LHCs in terms of protein structure and pigment arrangement. I encourage the authors to provide a supplementary figure showing the superimposed result of the two LHCs. A high-resolution structure of Chlamydomonas PSI-LHCII (7DZ7) was reported last year. The structural model of Lhca2 in this structure is almost complete, and can be used for comparison.*

Response to R3-02) Thank you for the comment. We have prepared a new superposition between *CrLhca2* of 7DZ7 and Pc-frLHC and added it as the **Figs. 6a, b, and c** (see **Response to R3-04**).

R3-03) *In plants and other green algal PSI-LHCI complexes, four normal LHCI (with three TMHs) form a half moon-shaped belt that attaches to one side of the PSI core. It seems that three such LHCI belts or eleven normal LHCI might also be able to form a circle. I assume that Prasiola crispa also contains normal LHCI, so I would like to hear the author's comments on what is the significance of this green alga adopting Pc-frLHC (with four TMHs) to form such a ring structure?*

Response to R3-03) Thank you for this important comment. We have revised the discussion on the significance of Pc-frLHC's ring-shaped structure in the Discussion section (**lines 298-311**).

We believe the significance of the ring-shaped structure of Pc-frLHC is a formation of a Chl network constructed by Chl-Chl interactions at the interfaces of the subunits. The Chl network seems to contribute to two important functions of Pc-frLHC, that is formation of LWC and efficient excitation of exit site Chls (Chl611', 612' and 610') with uphill EET.

Briefly, our Pc-frLHC structure revealed that the ring-shaped structure of 11 subunits contributes to the formation of 11 Chl pentamers. Each of these pentamers is composed of Chls 603, 609, 708, 613', and 614' and seems to contribute to far-red light absorption. The ring-shaped arrangement of the subunits enables Chl708 to interact with Chl 614' of the adjacent subunit, resulting in the formation of the Chl pentamers: Chl603-609-708 from one subunit and Chl613'-614' from the other. It is noteworthy that the Chl603-609-708 trimer forms due to the existence of Chl708, which is unique in the Pc-frLHC. However, the situation of the other LHCI is different. Since there are few Chl-Chl interactions between the neighboring LHCI monomers in an LHCI halfmoon-shaped belt, excitation energy in LHCI is not actively transported to neighboring monomers but PSI. If LHCI subunits were assembled in a similar way to Pc-frLHC, interactions between subunits in LHCI would energetically connect Chl molecules (ex. Chl609–Chl611') in two adjacent subunits. However, these interactions do not form Chl multimers as found in Pc-frLHC due to the absence of Chl708 in LHCI.

R3-04) *Could the authors compare the structures of the Pc-frLHC undecamer and the normal LHCI belt to see what is the difference between them, and whether the fourth TMH of the Pc-frLHC contributes to these differences.*

Response to R3-04) Thank you for the constructive suggestion. We have added a figure showing a structural comparison between *Chlamydomonas* LHCI in the PSI complex and Pc-frLHC (**Fig. 6, lines 155-164**). The fourth TMH, helix-F, is not directly involved in the interactions between subunits and seems to contribute less to the differences.

R3-05) Line 286, I don't think that Glu is the ligand for Chl 606 in other LHCI, it does not axially ligate the central Mg atom of Chl 606 in these structures. According to the structure of spinach LHCII and others, Chl 606 may be coordinated by a water molecule.

Response to R3-05) Thank you for your comment. As you pointed out, the Mg ligand of Chl606 in LHCII (Gln 130 in 1rwt) and LHCI (Glu144 of Lhca4, Glu155 of Lhca2, Glu160 in Lhca3 of 5l8r) was previously characterized as a water molecule (Liu 2004, Qin 2015), and the conserved Gln/Glu was found in these studies to interact indirectly with the central Mg of Chl606 via the water molecule. We have revised the corresponding sentences (**lines 303-305**).

Reviewer #4

R4-01) The findings reported in this article are stunning, but they are not (yet) sold as such. The authors could consider hiring a scientific writer. Overall it should be explained more why this data shows an important step forward in the field of photosynthesis. The abstract could be improved a lot based on the summary of the results provided by Reviewer 2:

“The most noteworthy results of this article are:

a) The presence of circular (bacterio-like) light-harvesting complexes (LHCII), named Pc-frLHC.

b) The unique features of the PC-frLHC antenna that appears to be the key adaptation factor for the survivance of Prasciola crispa are the following:

- Undecameric 11-fold symmetry;

- Exclusive subunit's structure with unprecedented Chls arrangement capable of long-wavelength absorbance. Structurally it is manifested in the presence of 2 low-wavelength Chls' - LWC708 & LWC725. Such structure leads to 2-steps uphill EET needed for PSII activation with far-red light.

c) The origin of Pc-frLHC refers to the PSI-LHCI family rather than PSII-LHCII.”I have some issues with the way the data is described and/or interpreted:

Response to R4-01) Thank you for the constructive comment. We have revised the abstract in accordance with this suggestion (**lines 48-60**).

R4-02) Line 231-233 *“Since only Chl b and the two carotenoids are excited in this analysis, this result suggests that the absorbed energies of Chl b and the two carotenoids are exclusively transferred to the 730 nm emitting species through Chl a.” Please specify in which region only Chl b and Cars are excited. For the 400 nm to 550 nm region this statement is obviously not correct.*

Response to R4-02) Thank you for the comment. Since the 460 nm absorbance of Chl *a* is quite small, we can safely assume the initial excitation occurred only on Chl *b* and carotenoids. We have revised the sentence to express this more clearly (**lines 247-249**).

R4-03) *Line 245-247: “Importantly, the above model requires that LWC708 intrinsically emits stronger fluorescence than LWC725.” What is the definition of intrinsic fluorescence? From the M and M section I understand that you mean the emission intensity if the energy would be the same or if the temperature would be infinite. This is not very clear. However the outcome is still valid. Your analyses suggests that the pigment pool contributing to the F713 emission is 12 times larger than that contributing to the F730 nm band. This is only valid when the radiative rates of both pools is equal. Strongly coupled pigments are known to be super-radiative, so this should be considered.*

Response to R4-03) Thank you for the constructive comments. As the reviewer suggested, the intrinsic fluorescence intensity is that observed when the temperature would be infinite. We have added the sentence “The intrinsic fluorescence intensity was that observed when the temperature would be infinite.” to the Methods section (**lines 562-563**).

As the reviewer suggested, we should also consider the possible effect of the modified oscillator strengths of the excited state formed by strongly coupled pigments. In fact, the large I_{F713}/I_{F730} ratio would also be explained based on the enhanced/reduced oscillator strength of these Chl pools. To examine this possibility, we calculated the transition dipole moment of the two excited states formed by the strongly coupled Chl pairs. The results of the calculations are listed in an additional table, **Supplementary Table 4**. As the table shows, there is no Chl pair whose upper excited state has a satisfactorily enhanced transition dipole moment. Therefore, we can exclude the possibility that the modified oscillator strength of the delocalized excited state is the origin of the large I_{F713}/I_{F730} ratio. We have revised the Discussion section (**lines 329-339**).

R4-04) *Line 334-336: The three factors described here are hardly related. The statements “problems remain” sound very negative. However, the problems described are not so large. The function of carotenoids are most likely the same as in every LHC (light absorption, structure stabilisation, photoprotection). Unsuccessful assignment of Chl b is a pity, but no serious issue. The function of LWC725 is probably just to enhance the light absorption in the far-red region.*

Response to R4-04) Thank you for the comments. To avoid the negative connotation, we have removed the word "problems" and changed the expression (**line 363**).

As for the last sentence of the same paragraph, we have deleted the speculative description about the photoprotective role of LWC725 (as pointed out in **R4-05**) and added a

sentence to mention its role in expanding the spectral window of the antenna (**lines 368-369**, see **Responses to R4-05**).

R4-05) Line 339-342: *Point 1: Do you mean high probability to be excited after energy-transfer? The initial excitation probability will be low. Point 2: I don't understand why to authors speculate about a role in quenching why their data shows that uphill energy transfer is fast (Fig 4a).*

Response to R4-05)

Thank you for the comments. Regarding point 1, we meant that LWC725 will have a high excitation probability after the equilibration, as you suggested. We have revised the corresponding sentences (**lines 368-369**). As for point 2, about the uphill energy transfer, it should be noted that the time constant of 25 ps obtained by the observation in **Fig. 8a (Fig.4a in the previous version)** is given by $1/(k_{up}+k_{down})$, where $k_{up/down}$ is the rate constant of the uphill/downhill energy transfer. In any case, we have removed the speculative description about the role of quenching and added a sentence to describe the above explanation in the subsection “**Fluorescence measurement revealed two distinct red Chl a pools**” (**lines 234-236**).

R4-06) Line 358-359: *“In addition, loro seems to energetically connect an isolated Chl604 and the Chl a network at the stromal side.” Energy transfer from the Chl604 through loro to the other Chls seems highly unlikely. I believe Chl to Car energy transfer has only been reported in the context of quenching.*

Response to R4-06) Thank you for this important suggestion. We agree with your opinion and have deleted the corresponding sentence (**lines 385-386**).

R4-07) Line 1001, Fig 5: *The Chl b + Car to Bulk Chl arrow should be broader to show very fast energy transfer. The blue arrow in the LWC708 band is not described in the text. The dashed arrow should be broader than the ~25 ps arrow as it would otherwise not be very competitive with the transfer to the LWC708, at least when this describes the transfer at physiological relevant conditions (this info should be added as some green uphill-arrows are shown, of which the rate is of course temperature dependent). The emission of the LWC725 band also occurs on the ns time-scale, this could be added.*

Response to R4-07) We have revised **Fig. 9 (Fig.5 in the previous manuscript)** according to the above comment and added the following to the legend of **Fig. 9 (lines 920-922)**: “The rate of the uphill energy transfer is temperature dependent. The winding blue arrow shows the energy migrations between the isoenergetic LWC₇₀₈ pigments.”.

Minor issues:

R4-08) *Line 50-52 “Cells of deeper layer escape from photodamage at the sacrifice of photosynthetically active radiation except infrared.” The sentence is hard to follow and infrared light is usually not considered as photosynthetically active radiation for green algae.*

R4-09) *Line 52-53 “At deeper layers, P. crispa achieves effective photosynthesis by low energy far-red light for photosystem II (PSII) excitation.” by being able to use far-red ...*

Response to R4-08, 09) Thank you for the comments. We have fully revised the abstract (see **Response to R4-01**).

R4-10) *Line 73-74: The LWC in LHCI are only 2 Chls, I don't think that classifies as an aggregate of molecules.*

Response to R4-10) Thank you for the comment. We have revised the manuscript (**lines 75**).

R4-11) *Line 81-82: “The EET mechanism of the far-red light-capturing system is noteworthy from the perspective of energy yield.” What is meant with this sentence?*

Response to R4-11) Thank you for the comment. We have revised the manuscript. The present result is noteworthy since it unveiled the structural basis for the survival strategy in the ultimate environment based on the utilization of far-red light (**lines 101-103**).

R4-12) *Line 1034: “Shorter wavelengths than 400 nm of the excitation light were cut by long-pass filters.” Was this light present due to the present of the second harmonic?*

Response to R4-12) As the reviewer suggested, we had to reduce the stray light due to the second-order diffraction of the excitation light (<400 nm). To describe this situation more clearly, we have revised the corresponding sentence (**lines 15-16 in the figure legend of Supplementary Fig. 1**).

Reviewer #3 (Remarks to the Author):

The authors adequately responded to all my questions. The revised manuscript is clearly written and much improved. Therefore, I would like to recommend for publication in Nature Communications.

Reviewer #4 (Remarks to the Author):

The authors have addressed all points that I raised. I now recommend the manuscript for publication in Nat Comm.